# ViTAE: Vision Transformer Advanced by Exploring Intrinsic Inductive Bias

**Yufei Xu**[1*]        **Qiming Zhang**[1*]        **Jing Zhang**[1]        **Dacheng Tao**[2,1]

[1]The University of Sydney, Australia,
[2]JD Explore Academy, China

{yuxu7116,qzha2506}@uni.sydney.edu.au, jing.zhang1@sydney.edu.au, dacheng.tao@gmail.com

## Abstract

Transformers have shown great potential in various computer vision tasks owing to their strong capability in modeling long-range dependency using the self-attention mechanism. Nevertheless, vision transformers treat an image as 1D sequence of visual tokens, lacking an intrinsic inductive bias (IB) in modeling local visual structures and dealing with scale variance. Alternatively, they require large-scale training data and longer training schedules to learn the IB implicitly. In this paper, we propose a new **V**ision **T**ransformer **A**dvanced by **E**xploring intrinsic IB from convolutions, *i.e.*, *ViTAE*. Technically, ViTAE has several spatial pyramid reduction modules to downsample and embed the input image into tokens with rich multi-scale context by using multiple convolutions with different dilation rates. In this way, it acquires an intrinsic scale invariance IB and is able to learn robust feature representation for objects at various scales. Moreover, in each transformer layer, ViTAE has a convolution block in parallel to the multi-head self-attention module, whose features are fused and fed into the feed-forward network. Consequently, it has the intrinsic locality IB and is able to learn local features and global dependencies collaboratively. Experiments on ImageNet as well as downstream tasks prove the superiority of ViTAE over the baseline transformer and concurrent works. Source code and pretrained models will be available at code.

## 1  Introduction

Transformers [79, 17, 40, 14, 46, 61] have shown a domination trend in NLP studies owing to their strong ability in modeling long-range dependencies by the self-attention mechanism [67, 81, 51]. Such success and good properties of transformers has inspired following many works that apply them in various computer vision tasks [19, 100, 97, 80, 7]. Among them, ViT [19] is the pioneering pure transformer model that embeds images into a sequence of visual tokens and models the global dependencies among them with stacked transformer blocks. Although it achieves promising performance on image classification, it requires large-scale training data and a longer training schedule. One important reason is that ViT

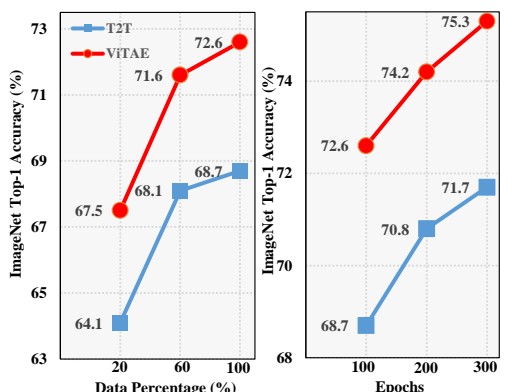

Figure 1: Comparison of data and training efficiency of T2T-ViT-7 and ViTAE-T on ImageNet.

*Equal Contribution. Interns at JD Explore Academy.

lacks intrinsic inductive bias (IB) in modeling local visual structures (*e.g.*, edges and corners) and dealing with objects at various scales like convolutions. Alternatively, ViT has to learn such IB implicitly from large-scale data.

Unlike vision transformers, Convolution Neural Networks (CNNs) naturally equip with the intrinsic IBs of scale-invariance and locality and still serve as prevalent backbones in vision tasks [26, 70, 62, 8, 96]. The success of CNNs inspires us to explore intrinsic IBs in vision transformers. We start by analyzing the above two IBs of CNNs, *i.e.*, locality and scale-invariance. Convolution that computes local correlation among neighbor pixels is good at extracting local features such as edges and corners. Consequently, CNNs can provide plentiful low-level features at the shallow layers [94], which are then aggregated into high-level features progressively by a bulk of sequential convolutions [32, 68, 71]. Moreover, CNNs have a hierarchy structure to extract multi-scale features at different layers [68, 38, 26]. Besides, intra-layer convolutions can also learn features at different scales by varying their kernel sizes and dilation rates [25, 70, 8, 45, 96]. Consequently, scale-invariant feature representation can be obtained via intra- or inter-layer feature fusion. Nevertheless, CNNs are not well suited to model long-range dependencies[2], which is the key advantage of transformers. An interesting question comes up: Can we improve vision transformers by leveraging the good properties of CNNs? Recently, DeiT [76] explores the idea of distilling knowledge from CNNs to transformers to facilitate training and improve the performance. However, it requires an off-the-shelf CNN model as the teacher and consumes extra training cost.

Different from DeiT, we explicitly introduce intrinsic IBs into vision transformers by re-designing the network structures in this paper. Current vision transformers always obtain tokens with single-scale context [19, 93, 80, 86, 47, 69, 77] and learn to adapt to objects at different scales from data. For example, T2T-ViT [93] improves ViT by delicately generating tokens in a soft split manner. Specifically, it uses a series of Tokens-to-Token transformation layers to aggregate single-scale neighboring contextual information and progressively structurizes the image to tokens. Motivated by the success of CNNs in dealing with scale variance, we explore a similar design in transformers, *i.e.*, intra-layer convolutions with different receptive fields [70, 91], to embed multi-scale context into tokens. Such a design allows tokens to carry useful features of objects at various scales, thereby naturally having the intrinsic scale-invariance IB and explicitly facilitating transformers to learn scale-invariant features more efficiently from data. On the other hand, low-level local features are fundamental elements to generate high-level discriminative features. Although transformers can also learn such features at shallow layers from data, they are not skilled as convolutions by design. Recently, [89, 43, 21] stack convolutions and attention layers sequentially and demonstrate that locality is a reasonable compensation of global dependency. However, this serial structure ignores the global context during locality modeling (and vice versa). To avoid such a dilemma, we follow the "divide-and-conquer" idea and propose to model locality and long-range dependencies in parallel and then fuse the features to account for both. In this way, we empower transformers to learn local and long-range features within each block more effectively.

Technically, we propose a new **Vi**sion **T**ransformers **A**dvanced by **E**xploring Intrinsic Inductive Bias (***ViTAE***), which is a combination of two types of basic cells, *i.e.*, reduction cell (RC) and normal cell (NC). RCs are used to downsample and embed the input images into tokens with rich multi-scale context while NCs aim to jointly model locality and global dependencies in the token sequence. Moreover, these two types of cells share a simple basic structure, *i.e.*, paralleled attention module and convolutional layers followed by a feed-forward network (FFN). It is noteworthy that RC has an extra pyramid reduction module with atrous convolutions of different dilation rates to embed multi-scale context into tokens. Following the setting in [93], we stack three reduction cells to reduce the spatial resolution by $1/16$ and a series of NCs to learn discriminative features from data. ViTAE outperforms representative vision transformers in terms of data efficiency and training efficiency (see Figure 1), as well as classification accuracy and generalization on downstream tasks.

Our contributions are threefold. **First**, we explore two types of intrinsic IB in transformers, *i.e.*, scale invariance and locality, and demonstrate the effectiveness of this idea in improving the feature learning ability of transformers. **Second**, we design a new transformer architecture named ViTAE based on two new reduction and normal cells to intrinsically incorporate the above two IBs. The proposed ViTAE embeds multi-scale context into tokens and learns both local and long-range features

---

[2]Despite projection in transformer can be viewed as $1 \times 1$ convolution [9], the term of convolution here refers to those with larger kernels, *e.g.*, $3 \times 3$, which are widely used in typical CNNs to extract spatial features.

effectively. **Third**, ViTAE outperforms representative vision transformers regarding classification accuracy, data efficiency, training efficiency, and generalization on downstream tasks. ViTAE achieves 75.3% and 82.0% top-1 accuracy on ImageNet with 4.8M and 23.6M parameters, respectively.

## 2 Related Work

### 2.1 CNNs with intrinsic IB

CNNs have led to a series of breakthroughs in image classification [38, 94, 26, 95, 87] and downstream computer vision tasks. The convolution operations in CNNs extract local features from the neighbor pixels within the receptive field determined by the kernel size [42]. Following the intuition that local pixels are more likely to be correlated in images [41], CNNs have the intrinsic IB in modeling locality. In addition to the locality, another critical topic in visual tasks is scale-invariance, where multi-scale features are needed to represent the objects at different scales effectively [49, 90]. For example, to effectively learn features of large objects, a large receptive field is needed by either using large convolution kernels [90, 91] or a series of convolution layers in deeper architectures [26, 32, 68, 71]. To construct multi-scale feature representation, the classical idea is using image pyramid [8, 1, 55, 4, 39, 16], where features are hand-crafted or learned from a pyramid of images at different resolutions respectively [44, 8, 52, 63, 35, 3]. Accordingly, features from the small scale image mainly encode the large objects while features from the large scale image respond more to small objects. In addition to the above inter-layer fusion way, another way is to aggregate multi-scale context by using multiple convolutions with different receptive fields within a single layer, *i.e.*, intra-layer fusion [96, 71, 70, 70, 72]. Either inter-layer fusion or intra-layer fusion empower CNNs an intrinsic IB in modeling scale-invariance. This paper introduces such an IB to vision transformers by following the intra-layer fusion idea and utilizing multiple convolutions with different dilation rates in the reduction cells to encode multi-scale context into each visual token.

### 2.2 Vision transformers with learned IB

ViT [19] is the pioneering work that applies a pure transformer to vision tasks and achieves promising results. However, since ViT lacks intrinsic inductive bias in modeling local visual structures, it indeed learns the IB from amounts of data implicitly. Following works along this direction are to simplify the model structures with fewer intrinsic IBs and directly learn them from large scale data [50, 74, 75, 22, 18, 20, 27] which have achieved promising results and been studied actively. Another direction is to leverage the intrinsic IB from CNNs to facilitate the training of vision transformers, *e.g.*, using less training data or shorter training schedules. For example, DeiT [76] proposes to distill knowledge from CNNs to transformers during training. However, it requires an off-the-shelf CNN model as a teacher, introducing extra computation cost during training. Recently, some works try to introduce the intrinsic IB of CNNs into vision transformers explicitly [23, 58, 21, 43, 15, 89, 83, 92, 6, 47, 11]. For example, [43, 21, 83] stack convolutions and attention layers sequentially, resulting in a serial structure and modeling the locality and global dependency accordingly. [80, 28] design sequential stage-wise structures while [47, 33] apply attention within local windows. However, these serial structure may ignore the global context during locality modeling (and vice versa). [88] establishes connection across different scales at the cost of heavy computation. Instead, we follow the "divide-and-conquer" idea and propose to model locality and global dependencies simultaneously via a parallel structure within each transformer layer. Conformer [58], the most relevant concurrent work to us, employs a unit to explore inter-block interactions between parallel convolution and transformer blocks. In contrast, in ViTAE, the convolution and attention modules are designed to be complementary to each other within the transformer block. In addition, Conformer is not designed to have inherent scale invariance IB.

## 3 Methodology

### 3.1 Revisit vision transformer

We first give a brief review of vision transformer in this part. To adapt transformers to vision tasks, ViT [19] first splits an image $x \in R^{H \times W \times C}$ into tokens with a reduction ratio of $p$ (*i.e.*, $x_t \in R^{((H \times W)/p^2) \times D}$), where $H$, $W$ and $C$ denote the height, width, and channel dimensions of

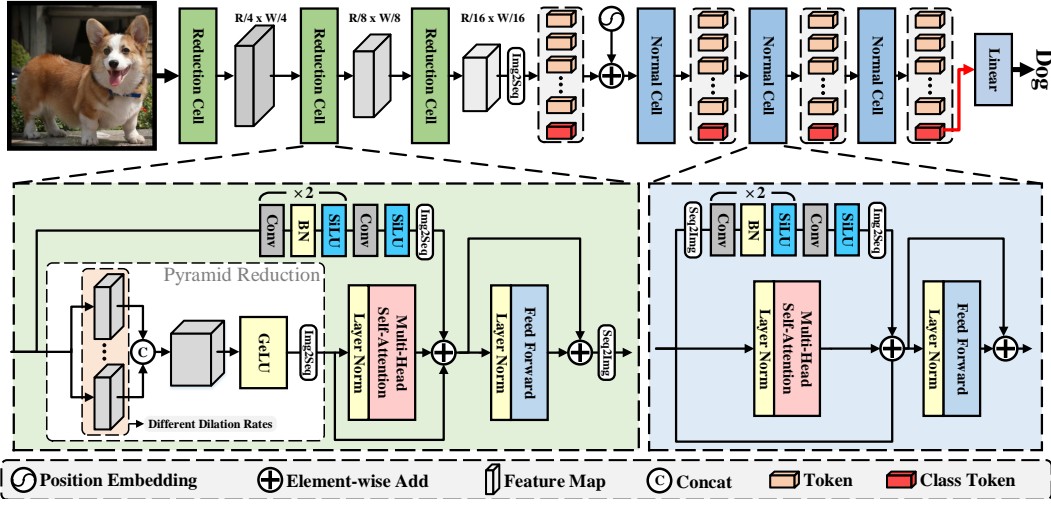

Figure 2: The structure of the proposed ViTAE. It is constructed by stacking three RCs and several NCs. Both types of cells share a simple basic structure, *i.e.*, an MHSA module and a parallel convolutional module followed by an FFN. In particular, RC has an extra pyramid reduction module using atrous convolutions with different dilation rates to embed multi-scale context into tokens.

the input image, $D = Cp^2$ denotes the token dimension. Then, an extra class token is concatenated to the visual tokens before adding position embeddings in an element-wise manner. The resulting tokens are fed into the following transformer layers. Each transformer layer is composed of two parts, *i.e.*, a multi-head self-attention module (MHSA) and a feed forward network (FFN).

**MHSA** Multi-head self-attention extends single-head self-attention (SHSA) by using different projection matrices for each head. Specifically, the input tokens $x_t$ are first projected to queries ($Q$), keys ($K$) and values ($V$) using projection matrices, *i.e.*, $Q, K, V = x_t W_Q, x_t Q_K, x_t Q_V$, where $W_{Q/K/V} \in R^{D \times D}$ denotes the projection matrix for query, key, and value, respectively. Then, the self-attention operation is calculated as:

$$Attention(Q, K, V) = softmax(\frac{QK^T}{\sqrt{D}})V. \tag{1}$$

This SHSA module is repeated for $h$ times to formulate the MHSA module, where $h$ is the number of heads. The output features of the $h$ heads are concatenated along the channel dimension and formulate the output of the MHSA module.

**FFN** FFN is placed on top of the MHSA module and applied to each token identically and separately. It consists of two linear transformations with an activation function in between. Besides, a layer normalization [2] and a shortcut are added before and aside from the MHSA and FFN, respectively.

### 3.2 Overview architecture of ViTAE

ViTAE aims to introduce the intrinsic IB in CNNs to vision transformers. As shown in Figure 2, ViTAE is composed of two types of cells, *i.e.*, RCs and NCs. RCs are responsible for embedding multi-scale context and local information into tokens, and NCs are used to further model the locality and long-range dependencies in the tokens. Taken an image $x \in R^{H \times W \times C}$ as input, three RCs are used to gradually downsample $x$ by $4\times$, $2\times$, and $2\times$, respectively. Thereby, the output tokens of the RCs are of size $[H/16, W/16, D]$ where $D$ is the token dimension (64 in our experiments). The output tokens of RCs are then flattened as $R^{HW/256 \times D}$, concatenated with the class token, and added by the sinusoid position encoding. Next, the tokens are fed into the following NCs, which keep the length of the tokens. Finally, the prediction probability is obtained using a linear classification layer on the class token from the last NC.

### 3.3 Reduction cell

Instead of directly splitting and flatten images into visual tokens based on a linear image patch embedding layer, we devise the reduction cell to embed multi-scale context and local information into visual tokens, which introduces the intrinsic scale-invariance and locality IBs from convolutions. Technically, RC has two parallel branches responsible for modeling locality and long-range dependency, respectively, followed by an FFN for feature transformation. We denote the input feature of the $i_{th}$ RC as $f_i \in R^{H_i \times W_i \times D_i}$. The input of the first RC is the image $x$. In the global dependencies branch, $f_i$ is firstly fed into a Pyramid Reduction Module (PRM) to extract multi-scale context, *i.e.*,

$$f_i^{ms} \triangleq PRM_i(f_i) = Cat([Conv_{ij}(f_i; s_{ij}, r_i)|s_{ij} \in \mathcal{S}_i, r_i \in \mathcal{R}]), \tag{2}$$

where $Conv_{ij}(\cdot)$ indicates the $j$th convolutional layer in the PRM ($PRM_i(\cdot)$). It uses a dilation rate $s_{ij}$ from the predefined dilation rate set $\mathcal{S}_i$ corresponding to the $i$th RC. Note that we use stride convolution to reduce the spatial dimension of features by a ratio $r_i$ from the predefined reduction ratio set $\mathcal{R}$. The conv features are concatenated along the channel dimension, *i.e.*, $f_i^{ms} \in R^{(W_i/p) \times (H_i/p) \times (|\mathcal{S}_i|D)}$, where $|\mathcal{S}_i|$ denotes the number of dilation rates in $\mathcal{S}_i$. $f_i^{ms}$ is then processed by an MHSA module to model long-range dependencies, *i.e.*,

$$f_i^g = MHSA_i(Img2Seq(f_i^{ms})), \tag{3}$$

where $Img2Seq(\cdot)$ is a simple reshape operation to flatten the feature map to a 1D sequence. In this way, $f_i^g$ embeds the multi-scale context in each token. In addition, we use a Parallel Convolutional Module (PCM) to embed local context within the tokens, which are fused with $f_i^g$ as follows:

$$f_i^{lg} = f_i^g + PCM_i(f_i). \tag{4}$$

Here, $PCM_i(\cdot)$ represents the PCM, which is composed of three stacked convolution layers and an $Img2Seq(\cdot)$ operation. It is noteworthy that the parallel convolution branch has the same spatial downsampling ratio as the PRM by using stride convolutions. In this way, the token features can carry both local and multi-scale context, implying that RC acquires the locality IB and scale-invariance IB by design. The fused tokens are then processed by the FFN, reshaped back to feature maps, and fed into the following RC or NC, *i.e.*,

$$f_{i+1} = Seq2Img(FFN_i(f_i^{lg}) + f_i^{lg}), \tag{5}$$

where the $Seq2Img(\cdot)$ is a simple reshape operation to reshape a token sequence back to feature maps. $FFN_i(\cdot)$ represents the FFN in the $i$th RC. In our ViTAE, three RCs are stacked sequentially to gradually reduce the input image's spatial dimension by $4\times$, $2\times$, and $2\times$, respectively. The feature maps generated by the last RC are of a size of $[H/16, W/16, D]$, which are then flattened into visual tokens and fed into the following NCs.

### 3.4 Normal cell

As shown in the bottom right part of Figure 2, NCs share a similar structure with the reduction cell except for the absence of the PRM. Due to the relatively small ($\frac{1}{16}\times$) spatial size of feature maps after RCs, it is unnecessary to use PRM in NCs. Given $f_3$ from the third RC, we first concatenate it with the class token $t_{cls}$, and then add it to the positional encodings to get the input tokens $t$ for the following NCs. Here we ignore the subscript for clarity since all NCs have an identical architecture but different learnable weights. $t_{cls}$ is randomly initialized at the start of training and fixed during the inference. Similar to the RC, the tokens are fed into the MHSA module, *i.e.*, $t_g = MHSA(t)$. Meanwhile, they are reshaped to 2D feature maps and fed into the PCM, *i.e.*, $t_l = Img2Seq(PCM(Seq2Img(t)))$. Note that the class token is discarded in PCM because it has no spatial connections with other visual tokens. To further reduce the parameters in NCs, we use group convolutions in PCM. The features from MHSA and PCM are then fused via element-wise sum, *i.e.*, $t_{lg} = t_g + t_l$. Finally, $t_{lg}$ are fed into the FFN to get the output features of NC, *i.e.*, $t_{nc} = FFN(t_{lg}) + t_{lg}$. Similar to ViT [19], we apply layer normalization to the class token generated by the last NC and feed it to the classification head to get the final classification result.

### 3.5 Model details

We use two variants of ViTAE in our experiments for a fair comparison of other models with similar model sizes. The details of them are summarized in Table 1. In the first RC, the default convolution kernel size is $7 \times 7$ with a

Table 1: Model details of two variants of ViTAE.

| Model | Reduction Cell | | Normal Cell | | | Params | Macs |
| | Dilation | Cells | Heads | Embed | Cells | (M) | (G) |
| --- | --- | --- | --- | --- | --- | --- | --- |
| ViTAE-T | $[1, 2, 3, 4] \downarrow$ | 3 | 4 | 256 | 7 | 4.8 | 1.5 |
| ViTAE-S | $[1, 2, 3, 4] \downarrow$ | 3 | 6 | 384 | 14 | 23.6 | 5.6 |

stride of 4 and dilation rates of $\mathcal{S}_1 = [1, 2, 3, 4]$. In the following two RCs, the convolution kernel size is $3 \times 3$ with a stride of 2 and dilation rates of $\mathcal{S}_2 = [1, 2, 3]$ and $\mathcal{S}_3 = [1, 2]$, respectively. Since the spatial dimension of tokens decreases, there is no need to use large kernels and dilation rates. PCM in both RCs and NCs comprises three convolutional layers with a kernel size of $3 \times 3$.

## 4 Experiments

### 4.1 Implementation details

We train and test the proposed ViTAE model on the standard ImageNet [38] dataset, which contains about 1.3 million images and covers 1k classes. Unless explicitly stated, the image size during training is set to $224 \times 224$. We use the AdamW [48] optimizer with the cosine learning rate scheduler and uses the data augmentation strategy exactly the same as T2T [93] for a fair comparison, regarding the training strategies and the size of models. We use a batch size of 512 for training all our models and set the initial learning rate to be 5e-4. The results of our models can be found in Table 2, where all the models are trained for 300 epochs on 8 V100 GPUs. The models are built on PyTorch [57] and TIMM [82].

### 4.2 Comparison with the state-of-the-art

We compare our ViTAE with both CNN models and vision transformers with similar model sizes in Table 2. Both Top-1/5 accuracy and real Top-1 accuracy on the ImageNet validation set are reported. We categorize the methods into CNN models, vision transformers with learned IB, and vision transformers with introduced intrinsic IB. Compared with CNN models, our ViTAE-T achieves a 75.3% Top-1 accuracy, which is better than ResNet-18 with more parameters. The real Top-1 accuracy of the ViTAE model is 82.9%, which is comparable to ResNet-50 that has four more times of parameters than ours. Similarly, our ViTAE-S achieves 82.0% Top-1 accuracy with half of the parameters of ResNet-101 and ResNet-152, showing the superiority of learning both local and long-range features from specific structures with corresponding intrinsic IBs by design. Similar phenomena can also be observed when comparing ViTAE-T with MobileNetV1 [31] and MobileNetV2 [65], where ViTAE obtains better performance with fewer parameters. When compared with larger models which are searched according to NAS [73], our ViTAE-S achieves a similar performance when using $384 \times 384$ images as input, which further shows the potential of vision transformers with intrinsic IB.

In addition, among the transformers with learned IB, ViT is the first pure transformer model for visual recognition. DeiT shares the same structure with ViT but uses different data augmentation and training strategies to facilitate the learning of transformers. DeiT⚗ denotes using an off-the-shelf CNN model as the teacher model to train DeiT, which introduces the intrinsic IB from CNN to transformer implicitly in a knowledge distillation manner, showing better performance than the vanilla ViT on the ImageNet dataset. It is exciting to see that our ViTAE-T with fewer parameters even outperforms the distilled model DeiT⚗, demonstrating the efficacy of introducing intrinsic IBs in transformers by design. Besides, compared with other transformers with explicit intrinsic IB, our ViTAE with fewer parameters also achieves comparable or better performance. For instance, ViTAE-T achieves comparable performance with LocalVit-T but has 1M fewer parameters, demonstrating the superiority of the proposed RCs and NCs in introducing intrinsic IBs.

### 4.3 Ablation study

We use T2T-ViT [93] as our baseline model in the following ablation study of our ViTAE. As shown in Table 3, we investigate the hyper-parameter settings in RCs and NCs by isolating them separately. All the models are trained for 100 epochs on ImageNet and follow the same training setting and data augmentation strategy as described in Section 4.1.

Table 2: Comparison of ViTAE and SOTA methods on the ImageNet validation set.

| Type | Model | Params (M) | MACs (G) | Input Size | ImageNet Top-1 | ImageNet Top-5 | Real Top-1 |
|---|---|---|---|---|---|---|---|
| CNN | ResNet-18 [26] | 11.7 | 3.6 | 224 | 70.3 | 86.7 | 77.3 |
| | ResNet-50 [26] | 25.6 | 7.6 | 224 | 76.7 | 93.3 | 82.5 |
| | ResNet-101 [26] | 44.5 | 15.2 | 224 | 78.3 | 94.1 | 83.7 |
| | ResNet-152 [26] | 60.2 | 22.6 | 224 | 78.9 | 94.4 | 84.1 |
| | EfficientNet-B0 [73] | 5.3 | 0.8 | 224 | 77.1 | 93.3 | 83.5 |
| | EfficientNet-B4 [73] | 19.3 | 8.4 | 380 | 82.9 | 96.4 | 88.0 |
| | MobileNetV1 [31] | 4.3 | 0.6 | 224 | 72.3 | - | - |
| | MobileNetV2(1.4) [65] | 6.9 | 0.6 | 224 | 74.7 | - | - |
| | RegNetY-600M [62] | 6.1 | 1.2 | 224 | 75.5 | - | - |
| | RegNetY-4GF [62] | 20.6 | 8.0 | 224 | 80.0 | - | 86.4 |
| | RegNetY-8GF [62] | 39.2 | 16.0 | 224 | 81.7 | - | 87.4 |
| Transformer | DeiT-T [76] | 5.7 | 2.6 | 224 | 72.2 | 91.1 | 80.6 |
| | DeiT-T⚗ [76] | 5.7 | 2.6 | 224 | 74.5 | 91.9 | 82.1 |
| | LocalViT-T [43] | 5.9 | 2.6 | 224 | 74.8 | 92.6 | - |
| | LocalViT-T2T [43] | 4.3 | 2.4 | 224 | 72.5 | - | - |
| | ConT-Ti [89] | 5.8 | 1.6 | 224 | 74.9 | - | - |
| | PiT-Ti [29] | 4.9 | 1.4 | 224 | 73.0 | - | - |
| | T2T-ViT-7 [93] | 4.3 | 1.2 | 224 | 71.7 | 90.9 | 79.7 |
| | **ViTAE-T** | 4.8 | 1.5 | 224 | 75.3 | 92.7 | 82.9 |
| | **ViTAE-T ↑ 384** | 4.8 | 5.7 | 384 | 77.2 | 93.8 | 84.4 |
| | CeiT-T [92] | 6.4 | 2.4 | 224 | 76.4 | 93.4 | 83.6 |
| | ConViT-Ti [15] | 6.0 | 2.0 | 224 | 73.1 | - | - |
| | CrossViT-Ti [6] | 6.9 | 3.2 | 224 | 73.4 | - | - |
| | **ViTAE-6M** | 6.5 | 2.0 | 224 | 77.9 | 94.1 | 84.9 |
| | PVT-T [80] | 13.2 | 3.8 | 224 | 75.1 | - | - |
| | LocalViT-PVT [43] | 13.5 | 9.6 | 224 | 78.2 | 94.2 | - |
| | ConViT-Ti+ [15] | 10.0 | 4.0 | 224 | 76.7 | - | - |
| | PiT-XS [29] | 10.6 | 2.8 | 224 | 78.1 | - | - |
| | ConT-M [89] | 19.2 | 6.2 | 224 | 80.2 | - | - |
| | **ViTAE-13M** | 13.2 | 3.4 | 224 | 81.0 | 95.4 | 86.8 |
| | DeiT-S [76] | 22.1 | 9.8 | 224 | 79.9 | 95.0 | 85.7 |
| | DeiT-S⚗ [76] | 22.1 | 9.8 | 224 | 81.2 | 95.4 | 86.8 |
| | PVT-S [80] | 24.5 | 7.6 | 224 | 79.8 | - | |
| | Conformer-Ti [58] | 23.5 | 5.2 | 224 | 81.3 | - | - |
| | Swin-T [47] | 29.0 | 9.0 | 224 | 81.3 | - | - |
| | CeiT-S [92] | 24.2 | 9.0 | 224 | 82.0 | 95.9 | 87.3 |
| | CvT-13 [83] | 20.0 | 9.0 | 224 | 81.6 | - | 86.7 |
| | ConViT-S [15] | 27.0 | 10.8 | 224 | 81.3 | - | - |
| | CrossViT-S [6] | 26.7 | 11.2 | 224 | 81.0 | - | - |
| | PiT-S [29] | 23.5 | 4.8 | 224 | 80.9 | - | - |
| | TNT-S [23] | 23.8 | 10.4 | 224 | 81.3 | 95.6 | - |
| | Twins-PCPVT-S[10] | 24.1 | 7.4 | 224 | 81.2 | - | - |
| | Twins-SVT-S [10] | 24.0 | 5.6 | 224 | 81.7 | - | - |
| | T2T-ViT-14 [93] | 21.5 | 5.2 | 224 | 81.5 | 95.7 | 86.8 |
| | **ViTAE-S** | 23.6 | 5.6 | 224 | 82.0 | 95.9 | 87.0 |
| | **ViTAE-S ↑ 384** | 23.6 | 20.2 | 384 | 83.0 | 96.2 | 87.5 |

We use ✓ and × to denote whether or not the corresponding module is enabled during the experiments. If all columns under the RC and NC are marked × as shown in the first row, the model becomes the standard T2T-ViT model. "Pre" indicates the output features of PCM and MHSA are fused before FFN while "Post" indicates a late fusion strategy correspondingly. "BN" indicates whether PCM uses BN after the convolutional layer or not. "×3" in the first column denotes that the dilation rate set is the same in the three RCs. "$[1, 2, 3, 4] \downarrow$" denotes using lower dilation rates in deeper RCs, *i.e.*, $\mathcal{S}_1 = [1, 2, 3, 4], \mathcal{S}_2 = [1, 2, 3], \mathcal{S}_3 = [1, 2]$.

As can be seen, using a pre-fusion strategy and BN achieves the best 69.9% Top-1 accuracy among other settings. It is noteworthy that all the variants of NC outperform the vanilla T2T-ViT, implying the effectiveness of PCM, which introduces the intrinsic locality IB in transformers. It can also be observed that BN plays an important role in improving the model's performance as it can help to alleviate the scale deviation between convolution's and attention's features. For the RC, we first investigate the impact of using different dilation rates in the PRM, as shown in the first column. As can be seen, using larger dilation rates (*e.g.*, 4 or 5) does not deliver better performance. We suspect that larger dilation rates may lead to plain features in the deeper RCs due to the smaller resolution of feature maps. To

Table 3: Ablation Study of RCs and NCs in our ViTAE. "Pre" indicates the output features of PCM and MHSA are fused before FFN while "Post" indicates a late fusion strategy correspondingly. "BN" indicates whether PCM uses BN or not. "$[1, 2, 3, 4] \downarrow$" denotes using smaller dilation rates in deeper RCs, *i.e.*, $\mathcal{S}_1 = [1, 2, 3, 4], \mathcal{S}_2 = [1, 2, 3], \mathcal{S}_3 = [1, 2]$.

| Reduction Cell | | Normal Cell | | | |
|---|---|---|---|---|---|
| Dilation ($\mathcal{S}_1 \sim \mathcal{S}_3$) | PCM | Pre | Post | BN | Top-1 |
| × | × | × | × | × | 68.7 |
| × | × | ✓ | × | × | 69.1 |
| × | × | × | ✓ | × | 69.0 |
| × | × | × | ✓ | ✓ | 68.8 |
| × | × | ✓ | × | ✓ | 69.9 |
| $[1, 2] \times 3$ | × | × | × | × | 69.5 |
| $[1, 2, 3] \times 3$ | × | × | × | × | 69.9 |
| $[1, 2, 3, 4] \times 3$ | × | × | × | × | 69.2 |
| $[1, 2, 3, 4, 5] \times 3$ | × | × | × | × | 68.9 |
| $[1, 2, 3, 4] \downarrow$ | × | × | × | × | 69.8 |
| $[1, 2, 3, 4] \downarrow$ | ✓ | × | × | × | 71.7 |
| $[1, 2, 3, 4] \downarrow$ | ✓ | ✓ | × | ✓ | 72.6 |

validate the hypothesis, we use smaller dilation rates in deeper RCs as denoted by $[1, 2, 3, 4] \downarrow$. As can be seen, it achieves comparable performance as $[1, 2, 3] \times$. However, compared with $[1, 2, 3, 4] \downarrow$, $[1, 2, 3] \times$ increases the amount of parameters from 4.35M to 4.6M. Therefore, we select $[1, 2, 3, 4] \downarrow$ as the default setting. In addition, after using PCM in the RC, it introduces the intrinsic locality IB, and the performance increases to 71.7% Top-1 accuracy. Finally, the combination of RCs and NCs achieves the best accuracy at 72.6%, demonstrating the complementarity between our RCs and NCs.

## 4.4  Data efficiency and training efficiency

To validate the effectiveness of the introduced intrinsic IBs in improving data efficiency and training efficiency, we compare our ViTAE with T2T-ViT at different training settings: (a) training them using 20%, 60%, and 100% ImageNet training set for equivalent 100 epochs on the full ImageNet training set, *e.g.*, we employ 5 times epochs when using 20% data for training compared with using 100% data; and (b) training them using the full ImageNet training set for 100, 200, and 300 epochs respectively. The results are shown in Figure 1. As can be seen, ViTAE consistently outperforms the T2T-ViT baseline by a large margin in terms of both data efficiency and training efficiency. For example, ViTAE using only 20% training data achieves comparable performance with T2T-ViT using all data. When 60% training data are used, ViTAE significantly outperforms T2T-ViT using all data by about an absolute 3% accuracy. It is also noteworthy that ViTAE trained for only 100 epochs has outperformed T2T-ViT trained for 300 epochs. After training ViTAE for 300 epochs, its performance is significantly boosted to 75.3% Top-1 accuracy. With the proposed RCs and NCs, the transformer layers in our ViTAE only need to focus on modeling long-range dependencies, leaving the locality and multi-scale context modeling to its convolution counterparts, *i.e.*, PCM and PRM. Such a "divide-and-conquer" strategy facilitates the training of vision transformers, making it possible to learn more efficiently with less training data and fewer training epochs.

To further validate the data efficiency of ViTAE model, we train the ViTAE model from scratch on the smaller datasets, *i.e.*, Cifar10 and Cifar100. The results are summarized in Table 4. It can be viewed that with only 1/7 number of epochs, the ViTAE-T model achieves better classification

performance on Cifar10 dataset, with far fewer parameters (4.8M v.s. 86M), which further confirms ViTAE model's data efficiency.

Table 4: Results of training from scratch on Cifar10/100.

| Model | Params (M) | Top-1 Acc | Epochs | Dataset |
|---|---|---|---|---|
| DeiT-B | 86.0 | 97.5 | 7000 | Cifar10 |
| ViTAE-T | 4.8 | 97.7 | 1000 | Cifar10 |
| ViTAE-T | 4.8 | 85.0 | 1000 | Cifar100 |

## 4.5  Generalization on downstream tasks

Table 5: Generalization of ViTAE and SOTA methods on different downstream tasks.

| Model | Params (M) | Cifar10 | Cifar100 | iNat19 | Cars | Flowers | Pets |
|---|---|---|---|---|---|---|---|
| Grafit ResNet-50 [78] | 25.6 | - | - | 75.9 | 92.5 | 98.2 | - |
| EfficientNet-B5 [73] | 30 | 98.1 | 91.1 | - | - | 98.5 | - |
| ViT-B/16 [19] | 86.5 | 98.1 | 87.1 | - | - | 89.5 | 93.8 |
| ViT-L/16 [19] | 304.3 | 97.9 | 86.4 | - | - | 89.7 | 93.6 |
| DeiT-B [76] | 86.6 | 99.1 | 90.8 | 77.7 | 92.1 | 98.4 | - |
| T2T-ViT-14 [93] | 21.5 | 98.3 | 88.4 | - | - | - | - |
| ViTAE-T | 4.8 | 97.3 | 86.0 | 73.3 | 89.5 | 97.5 | 92.6 |
| ViTAE-S | 23.6 | 98.8 | 90.8 | 76.0 | 91.4 | 97.8 | 94.2 |

We further investigate the generalization of the proposed ViTAE models on downstream tasks by fine-tuning them on the training sets of several fine-grained classification tasks[3], including Flowers [53], Cars [36], Pets [56], and iNaturalist19. We also fine-tune the proposed ViTAE models on Cifar10 [37] and Cifar100 [37]. The results are shown in Table 5. It can be seen that ViTAE achieves SOTA performance on most of the datasets using comparable or fewer parameters. These results demonstrate that the good generalization ability of our ViTAE.

## 4.6  Visual inspection of ViTAE

To further analyze the property of our ViTAE, we first calculate the average attention distance of each layer in ViTAE-T and the baseline T2T-ViT-7 on the ImageNet test set, respectively. The results are shown in Figure 3. It can be observed that with the usage of PCM, which focuses on modeling locality, the transformer layers in the proposed NCs can better focus on modeling long-range dependencies, especially in shallow layers. In the deep layers, the average attention distances of ViTAE-T and T2T-ViT-7 are almost the same since modeling long-range dependencies is much more important. These results confirm the effectiveness of the adopted "divide-and-conquer" idea in the proposed ViTAE, *i.e.*, introducing the intrinsic locality IB from convolutions into vision transformers makes it possible that transformer layers only need to be responsible to long-range dependencies, since locality can be well modeled by convolutions in PCM.

Besides, we apply Grad-CAM [66] on the MHSA's output in the last NC to qualitatively inspect ViTAE. The visualization results are provided in Figure 4. Compared with the baseline T2T-ViT, our ViTAE covers the single or multiple targets in the images

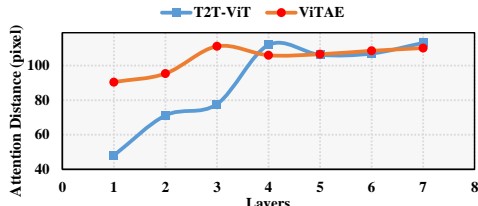

Figure 3: The average per-layer attention distance of T2T-ViT-7 and our ViTAE-T.

more precisely and attends less to the background. Moreover, ViTAE can better handle the scale variance issue as shown in Figure 4(b). Namely, it can precisely cover the birds no matter they are in small, middle, or large size. Such observations demonstrate that introducing the intrinsic IBs of locality and scale-invariance from convolutions to transformers helps ViTAE learn more discriminate features than the pure transformers.

---

[3]The performance of ViTAE on dense prediction tasks such as detection, segmentation, pose estimation, can be found in the supplementary material.

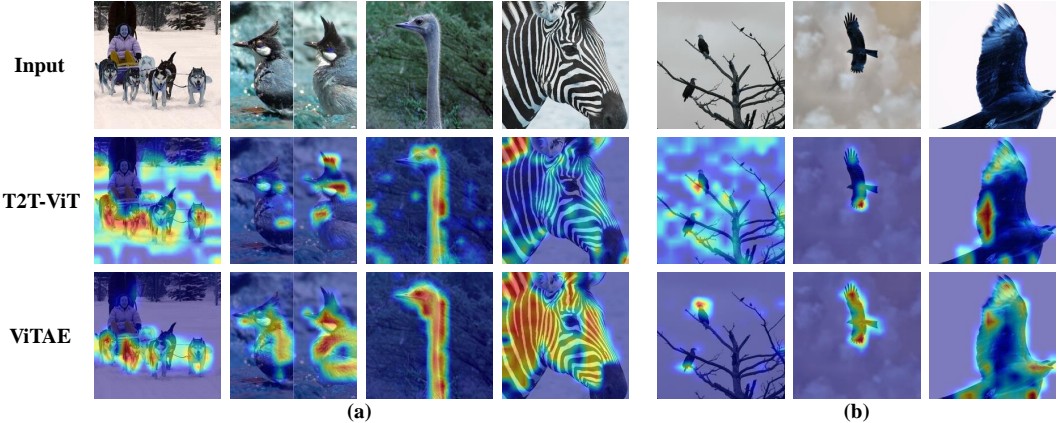

Figure 4: Visual inspection of T2T-ViT and ViTAE using Grad-CAM [66]. (a) Images containing multiple or single objects and the heatmaps. (b) Images containing the same class of objects at different scales and the heatmaps (Best viewed in color).

## 5 Limitation and discussion

In this paper, we explore two types of IBs and incorporate them into transformers through the proposed reduction and normal cells. With the collaboration of these two cells, our ViTAE model achieves impressive performance on the ImageNet with fast convergence and high data efficiency. Nevertheless, due to computational resource constraints, we have not scaled the ViTAE model and train it on large-size dataset, *e.g.*, ImageNet-21K [38] and JFT-300M [30]. Although it remains unclear by now, we are optimistic about its scale property from the following preliminary evidence. As illustrated in Figure 2, our ViTAE model can be viewed as an intra-cell ensemble of complementary transformer layers and convolution layers owing to the skip connection and parallel structure. According to the attention distance analysis shown in Figure 3, the ensemble nature enables the transformer layers and convolution layers to focus on what they are good at, *i.e.*, modeling long-range dependencies and locality. Therefore, ViTAE is very likely to learn better feature representation from large-scale data. Besides, we only study two typical IBs in this paper. More kinds of IBs such as constituting viewpoint invariance [64] can be explored in the future study.

## 6 Conclusion

In this paper, we re-design the transformer block by proposing two basic cells (reduction cells and normal cells) to incorporate two types of intrinsic inductive bias (IB) into transformers, *i.e.*, locality and scale-invariance, resulting in a simple yet effective vision transformer architecture named ViTAE. Extensive experiments show that ViTAE outperforms representative vision transformers in various respects including classification accuracy, data efficiency, training efficiency, and generalization ability on downstream tasks. We plan to scale ViTAE to the large or huge model size and train it on large-size datasets in the future study. In addition, other kinds of IBs will also be investigated. We hope that this study will provide valuable insights to the following studies of introducing intrinsic IB into vision transformers and understanding the impact of intrinsic and learned IBs.

**Acknowledgement** Dr. Jing Zhang is supported by the ARC project FL-170100117.

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
