# A Appendix

## A.1 Results of other ViTAE variants

To make a fair comparison of our ViTAE model and other methods, we further design three more ViTAE variants and present their results in Table 6. As can be seen, our ViTAE-T model achieves 75.3% Top-1 accuracy on ImageNet [37] with 4.8M parameters, which outperforms other transformer methods with even more than 5M parameters. With 6.5M parameters, ViTAE obtains 77.9% Top-1 accuracy, outperforming ResNet-50 [26] with 1.2% absolute improvement and 3/4 less parameters. These results demonstrate the potential of transformers with intrinsic IBs. Among vision transformers models, ViTAE-T outperforms DeiT-T🏫 [75] with similar parameters, while ViTAE-T does not require extra teacher models. Similarly, with 6.5M parameters, ViTAE-6M outperforms both transformers with learned IB [15] and transformers with intrinsic IB in a serial manner [91]. Similar phenomena can also be observed with the size of models increase, *e.g.*, the ViTAE-S model achieves state-of-the-art performance with fewer parameters.

Besides, the classic vision transformer design is not well suited for downstream tasks like detection, segmentation, pose estimation and *etc*. The stage-wise design can better adapt to the popular vision backbones for these tasks. To fully explore the potential of the proposed RC and NC modules, we also design the stage-wise variants of ViTAE model as shown in Table 7 and their classification performances are summarized in Table 6. The "NC Arrangement" means the number of NCs arranged after each RC. We follow the ResNet [26] and Swin [46] experience to design ViTAE's stage variants, where the spatial size are downsampled by $4, 2, 2, 2$ in each stage, except for the ViTAE-T-Stage model, where we only adopts the first three stages since the network is shallow. As shown in Table 6, the stage-wise design can further improve the performance with fewer parameters.

## A.2 Performance on downstream tasks

We further validate the performance of the proposed ViTAE models on detection, segmentation, pose estimation, and video object segmentation.

## A.3 Object detection

To evaluate ViTAE's performance on object detection and instance segmentation tasks, we adopt Mask RCNN [24] and Cascade RCNN [5] as the detection framework, and finetune the models on COCO 2017 dataset, which contains 118K training, 5K validation and 20K test-dev images. We adopt exactly the same training setting used in Swin [46], *i.e.*, multi-scale training and AdamW optimizer. We compare the performance of ViTAE with the classic CNN backbone, *i.e.*, ResNet [26], and the transformer structure, *i.e.*, Swin [46]. The comparisons are conducted by simply replacing the backbone while keeping other configurations unchanged. The results are summarized in Table 8. It can be concluded that the ViTAE-S-Stage model can obtain the best performance on object detection and instance segmentation, with both frameworks.

## A.4 Semantic segmentation

We evaluate the semantic segmentation performance of the ViTAE model on ADE20K [97, 98]. The ADE20K dataset covers 150 semantic categories with 20K images for training and 2K images for validation. We follow Swin's [46] training and testing setting. We adopt UperNet [84] as the segmentation framework and train the models for 160K iterations, with default setting used in mmsegmentation [12]. The results can be found in Table 8. It can be concluded that the ViTAE backbone using 10M fewer parameters achieves better performance than ResNet-50 [26] and Swin-T [46] on segmentation.

## A.5 Pose estimation

For human pose estimation, we adopt the simple baseline [83] as the pose estimation framework and test the ViTAE models' performance on the COCO dataset. The experiment are conducted following the default settings used in mmpose [13]. As shown in Table 8, the ViTAE-based model obtains an absolute 2% mAP gain than ResNet [26] models with 7M parameters fewer.

## A.6 Video object segmentation

For VOS tasks, the STM [53] framework is adopted and we replace the backbone network with the ViTAE-T-Stage model. Davis-2016 [58] and Davis-2017 [59] are used as the benchmark datasets. The first dataset contains 20 videos annotated with masks each for a single target object. The Davis-2017 dataset is a multi-object extension of Davis-2016, with 59 objects in 30 videos. The training and testing setting are the same as in STM [53]. With 29M parameters fewer, the ViTAE-based STM achieves an absolute 0.5 J&F scores improvement on Davis 2016 and 0.7 J&F scores improved on

Table 6: Comparison with SOTA methods.

| Type | Model | Params (M) | MACs (G) | Input Size | ImageNet Top-1 | Top-5 | Real Top-1 |
|---|---|---|---|---|---|---|---|
| CNN | ResNet-18 [26] | 11.7 | 3.6 | 224 | 70.3 | 86.7 | 77.3 |
| | ResNet-50 [26] | 25.6 | 7.6 | 224 | 76.7 | 93.3 | 82.5 |
| | ResNet-101 [26] | 44.5 | 15.2 | 224 | 78.3 | 94.1 | 83.7 |
| | ResNet-152 [26] | 60.2 | 22.6 | 224 | 78.9 | 94.4 | 84.1 |
| | EfficientNet-B0 [72] | 5.3 | 0.8 | 224 | 77.1 | 93.3 | 83.5 |
| | EfficientNet-B4 [72] | 19.3 | 8.4 | 380 | 82.9 | 96.4 | 88.0 |
| | MobileNetV1 [30] | 4.3 | 0.6 | 224 | 72.3 | - | - |
| | MobileNetV2(1.4) [64] | 6.9 | 0.6 | 224 | 74.7 | - | - |
| | RegNetY-600M [61] | 6.1 | 1.2 | 224 | 75.5 | - | - |
| | RegNetY-4GF [61] | 20.6 | 8.0 | 224 | 80.0 | - | 86.4 |
| | RegNetY-8GF [61] | 39.2 | 16.0 | 224 | 81.7 | - | 87.4 |
| Transformer | DeiT-T [75] | 5.7 | 2.6 | 224 | 72.2 | 91.1 | 80.6 |
| | DeiT-T⚗ [75] | 5.7 | 2.6 | 224 | 74.5 | 91.9 | 82.1 |
| | LocalViT-T [42] | 5.9 | 2.6 | 224 | 74.8 | 92.6 | - |
| | LocalViT-T2T [42] | 4.3 | 2.4 | 224 | 72.5 | - | - |
| | ConT-Ti [88] | 5.8 | 1.6 | 224 | 74.9 | - | - |
| | PiT-Ti [28] | 4.9 | 1.4 | 224 | 73.0 | - | - |
| | T2T-ViT-7 [92] | 4.3 | 1.2 | 224 | 71.7 | 90.9 | 79.7 |
| | **ViTAE-T** | 4.8 | 1.5 | 224 | 75.3 | 92.7 | 82.9 |
| | **ViTAE-T-Stage** | 4.8 | 2.3 | 224 | 76.8 | 93.5 | 84.0 |
| | CeiT-T [91] | 6.4 | 2.4 | 224 | 76.4 | 93.4 | 83.6 |
| | ConViT-Ti [15] | 6.0 | 2.0 | 224 | 73.1 | - | - |
| | CrossViT-Ti [6] | 6.9 | 3.2 | 224 | 73.4 | - | - |
| | **ViTAE-6M** | 6.5 | 2.0 | 224 | 77.9 | 94.1 | 84.9 |
| | PVT-T [79] | 13.2 | 3.8 | 224 | 75.1 | - | - |
| | LocalViT-PVT [42] | 13.5 | 9.6 | 224 | 78.2 | 94.2 | - |
| | ConViT-Ti+ [15] | 10.0 | 4.0 | 224 | 76.7 | - | - |
| | PiT-XS [28] | 10.6 | 2.8 | 224 | 78.1 | - | - |
| | ConT-M [88] | 19.2 | 6.2 | 224 | 80.2 | - | - |
| | **ViTAE-13M** | 13.2 | 3.4 | 224 | 81.0 | 95.4 | 86.8 |
| | DeiT-S [75] | 22.1 | 9.8 | 224 | 79.9 | 95.0 | 85.7 |
| | DeiT-S⚗ [75] | 22.1 | 9.8 | 224 | 81.2 | 95.4 | 86.8 |
| | PVT-S [79] | 24.5 | 7.6 | 224 | 79.8 | - | - |
| | Conformer-Ti [57] | 23.5 | 5.2 | 224 | 81.3 | - | - |
| | Swin-T [46] | 29.0 | 9.0 | 224 | 81.3 | - | - |
| | CeiT-S [91] | 24.2 | 9.0 | 224 | 82.0 | 95.9 | 87.3 |
| | CvT-13 [82] | 20.0 | 9.0 | 224 | 81.6 | - | 86.7 |
| | ConViT-S [15] | 27.0 | 10.8 | 224 | 81.3 | - | - |
| | CrossViT-S [6] | 26.7 | 11.2 | 224 | 81.0 | - | - |
| | PiT-S [28] | 23.5 | 4.8 | 224 | 80.9 | - | - |
| | TNT-S [23] | 23.8 | 10.4 | 224 | 81.3 | 95.6 | - |
| | Twins-PCPVT-S[10] | 24.1 | 7.4 | 224 | 81.2 | - | - |
| | Twins-SVT-S [10] | 24.0 | 5.6 | 224 | 81.7 | - | - |
| | T2T-ViT-14 [92] | 21.5 | 5.2 | 224 | 81.5 | 95.7 | 86.8 |
| | **ViTAE-S** | 23.6 | 5.6 | 224 | 82.0 | 95.9 | 87.0 |
| | **ViTAE-S-Stage** | 19.2 | 6.0 | 224 | 82.2 | 96.0 | 87.4 |
| | ViT-B/16 [19] | 86.5 | 18.7 | 384 | 77.9 | - | - |
| | ViT-L/16 [19] | 304.3 | 65.8 | 384 | 76.5 | - | - |
| | DeiT-B [75] | 86.6 | 34.6 | 224 | 81.8 | 95.6 | 86.7 |
| | PVT-M [79] | 44.2 | 13.2 | 224 | 81.2 | - | - |
| | PVT-L [79] | 61.4 | 19.6 | 224 | 81.7 | - | - |
| | Conformer-S [57] | 37.7 | 10.6 | 224 | 83.4 | - | - |
| | Swin-S [46] | 50.0 | 17.4 | 224 | 83.0 | | |
| | ConT-B [88] | 39.6 | 12.8 | 224 | 81.8 | - | - |
| | CvT-21 [82] | 32.0 | 14.2 | 224 | 82.5 | - | 87.2 |
| | ConViT-S+ [15] | 48.0 | 20.0 | 224 | 82.2 | - | - |
| | ConViT-B [15] | 86.0 | 34.0 | 224 | 82.4 | - | - |
| | ConViT-B+ [15] | 152.0 | 60.0 | 224 | 82.5 | - | - |
| | PiT-B [28] | 73.8 | 25.0 | 224 | 82.0 | - | - |
| | TNT-B [23] | 65.6 | 28.2 | 224 | 82.8 | 96.3 | - |
| | T2T-ViT-19 [92] | 39.2 | 8.9 | 224 | 81.9 | 95.7 | 86.9 |
| | **ViTAE-B-Stage** | 48.5 | 13.8 | 224 | 83.6 | 96.4 | 87.9 |

Table 7: Model details of ViTAE variants.

| Model | Reduction Cell dilation | Reduction Cell cells | Normal Cell heads | Normal Cell embed | Normal Cell cells | NC Arrangement | Params (M) | Macs (G) |
|---|---|---|---|---|---|---|---|---|
| ViTAE-T | $[1, 2, 3, 4] \downarrow$ | 3 | 4 | 256 | 7 | 0, 0, 7 | 4.8 | 1.5 |
| ViTAE-6M | $[1, 2, 3, 4] \downarrow$ | 3 | 4 | 256 | 10 | 0, 0, 10 | 6.5 | 2.0 |
| ViTAE-13M | $[1, 2, 3, 4] \downarrow$ | 3 | 4 | 320 | 11 | 0, 0, 11 | 13.2 | 3.4 |
| ViTAE-S | $[1, 2, 3, 4] \downarrow$ | 3 | 6 | 384 | 14 | 0, 0, 14 | 23.6 | 5.6 |
| ViTAE-T-Stage | $[1, 2, 3, 4] \downarrow$ | 3 | 4 | 256 | 7 | 1, 1, 5 | 4.8 | 2.3 |
| ViTAE-S-Stage | $[1, 2, 3, 4] \downarrow$ | 4 | 8 | 512 | 14 | 1, 1, 11, 1 | 19.2 | 6.0 |
| ViTAE-B-Stage | $[1, 2, 3, 4] \downarrow$ | 4 | 8 | 768 | 17 | 1, 1, 14, 1 | 48.5 | 13.8 |

Table 8: ViTAE on downstream tasks.

| Detection-COCO | | | | | |
|---|---|---|---|---|---|
| Backbone | Method | Lr Schd | box mAP | mask mAP | params (M) |
| ResNet-50 [26] | Mask RCNN [24] | 1x | 38.2 | 34.7 | 44 |
| Swin-T [46] | Mask RCNN [24] | 1x | 43.7 | 39.8 | 48 |
| **ViTAE-S-Stage** | **Mask RCNN** [24] | **1x** | **44.6** | **40.2** | **37** |
| ResNet-50 [26] | Cascade RCNN [5] | 1x | 41.2 | 35.9 | 82 |
| Swin-T [46] | Cascade RCNN [5] | 1x | 48.1 | 41.7 | 86 |
| **ViTAE-S-Stage** | **Cascade RCNN** [5] | **1x** | **48.9** | **42.0** | **75** |
| Segmentation-ADE20K | | | | | |
| Backbone | Method | Lr Schd | mIoU | mIoU(ms+flip) | params (M) |
| Swin-T [46] | UPerNet [84] | 160K | 44.5 | 45.8 | 60 |
| **ViTAE-S-Stage** | **UPerNet** [84] | **160K** | **45.4** | **47.8** | **49** |
| Pose-COCO | | | | | |
| Backbone | Method | InputSize | mAP | mAR | params (M) |
| ResNet-50 [26] | SimpleBaseline [83] | 256x192 | 71.8 | 77.3 | 34 |
| **ViTAE-S-Stage** | **SimpleBaseline** [83] | **256x192** | **73.7** | **79.0** | **27** |
| VOS-Davis2017 | | | | | |
| Backbone | Method | J | F | J&F | params (M) |
| ResNet-50 [26] | STM [53] | 79.2 | 84.3 | 81.8 | 39 |
| **ViTAE-T-Stage** | **STM** [53] | **79.4** | **85.5** | **82.5** | **19** |
| VOS-Davis2016 | | | | | |
| Backbone | Method | J | F | J&F | params (M) |
| ResNet-50 [26] | STM [53] | 88.7 | 89.9 | 89.3 | 39 |
| **ViTAE-T-Stage** | **STM** [53] | **89.2** | **90.4** | **89.8** | **19** |

Davis 2017 dataset. It can be concluded that the intrinsic IB introduced by the RC and NC module indeed improves the generalization ability of backbone networks for various downstream tasks.

### A.7 More comparisons of data efficiency and training efficiency.

Besides T2T-ViT [92] for the evaluation of the data efficiency and training efficiency, we further train DeiT [75] with 20%, 60% and 100% data for 100 epochs and train it with 100% data for 100, 200, and 300 epochs. Its results can be viewed in Figure 5. It can be observed that, with inductive bias introduced, T2T-ViT achieves better performance with less data when compared with DeiT. Without loss of generality, T2T-ViT outperforms DeiT with fewer training epochs, *e.g.*, T2T-ViT with 20% data can perform comparably to DeiT with 100% data. With more intrinsic inductive bias introduced, ViTAE outperforms T2T-ViT with fewer data and fewer epochs. Such observation confirms that with proper intrinsic inductive bias, the training of transformer models can be both data efficiency and training efficiency.

### A.8 Analysis of position embedding

As CNN can encode position information with padding [11], we further disable the position em-

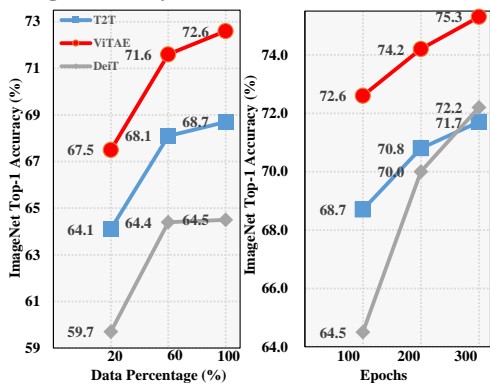

Figure 5: Comparisons of DeiT, T2T-ViT and ViTAE in terms of data efficiency and training efficiency on ImageNet.

Table 9: ViTAE with different PE.

| | Sinusoid | No | Learnable |
|---|---|---|---|
| Top-1 | 75.3 | 75.3 | 75.1 |

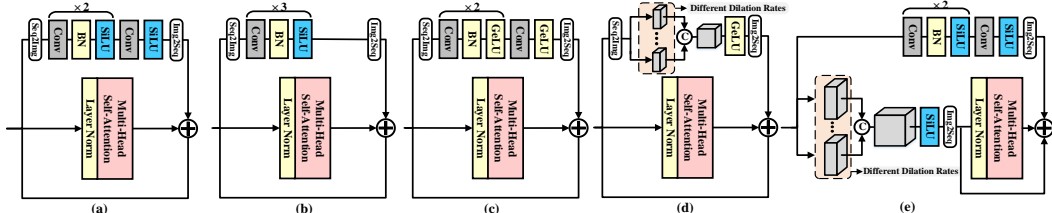

Figure 6: Different structures used in the ablation study. (a) Origin ViTAE structure. (b) Adding another BN in the PCM. (c) Replacing SiLU with GeLU in the PCM. (d) Replacing PCM with PRM. (e) Replacing GeLU with SiLU in the PRM.

bedding and train the ViTAE model without position embedding for 300 epochs. The results are summarized in Table 9. It can be seen that removing position embedding (PE) in the ViTAE model does not downgrade its performance. Such phenomena show that the PCM and PRM modules utilized in the ViTAE can aid the model in making sense of location information.

### A.9 More Ablation Studies

To further analyze the structure of our ViTAE model, we conduct more ablation studies related to the proposed reduction cells and normal cells. As shown in Table 10 and Figure 6, we first add another batch normalization [33] in the PCM mod-

Table 10: More ablation studies. (a), (b), (c), (d), (e) correspond to different structures in Figure 6.

|       | (a)  | (b)  | (c)  | (d)  | (e)  |
| ----- | ---- | ---- | ---- | ---- | ---- |
| Top-1 | 72.6 | 69.6 | 72.3 | 71.9 | 72.4 |

ule to make the PCM module has three exactly same convolution layers. However, such a structure downgrades the performance by 3%. As there is layer normalization [2] before the input to the FFN module, the combination of batch normalization and layer normalization may conflict with each other, resulting in performance degradation. Another design choice we tried is replacing the PCM with PRM modules (Figure 6 (d)) to introduce both scale-invariance and locality in the parallel branch. However, such a design shows a small drop in the performance, indicating that not only introducing inductive bias is important in transformers, but also the way in which these inductive biases are introduced also matters. What's more, we test the activation function used in the PCM and PRM modules. By default, we adopt SiLU in PCM, following the design choice in pioneering CNN networks and GeLu in PRM following previous transformers' design. By replacing the SiLU with GeLU (Figure 6 (c)), the performance drops a little. Similar phenomena can be observed when all PCM and PRM modules adopt SiLU as activation functions (Figure 6 (e)).

### A.10 More visual results.

We also provide more visual inspection results of ViTAE using Grad-CAM [65] in Figure 7 8 9 10 and compare it with T2T-ViT [92] in Figure 11. It can be seen that our ViTAE can cover the targets more precisely and compress the noise introduced by the complex background. Such phenomena confirm that with the introduced inductive bias, the ViTAE model can better adapt to targets in different situations and thus achieves better performance on the vision tasks.

| Input | ViTAE | Input | ViTAE | Input | ViTAE |
|-------|-------|-------|-------|-------|-------|

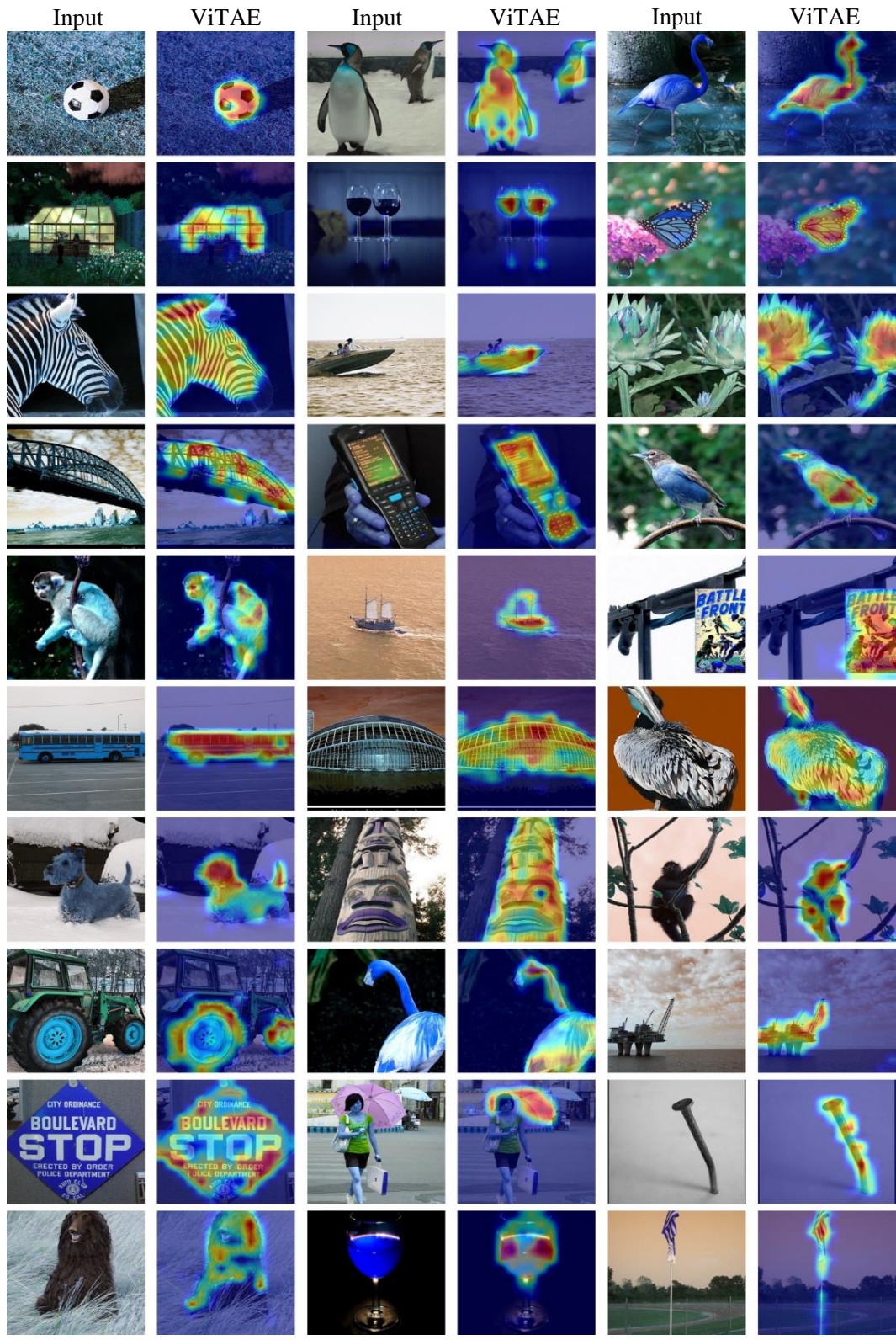

Figure 7: More visual results of ViTAE.

| Input | ViTAE | Input | ViTAE | Input | ViTAE |
|-------|-------|-------|-------|-------|-------|

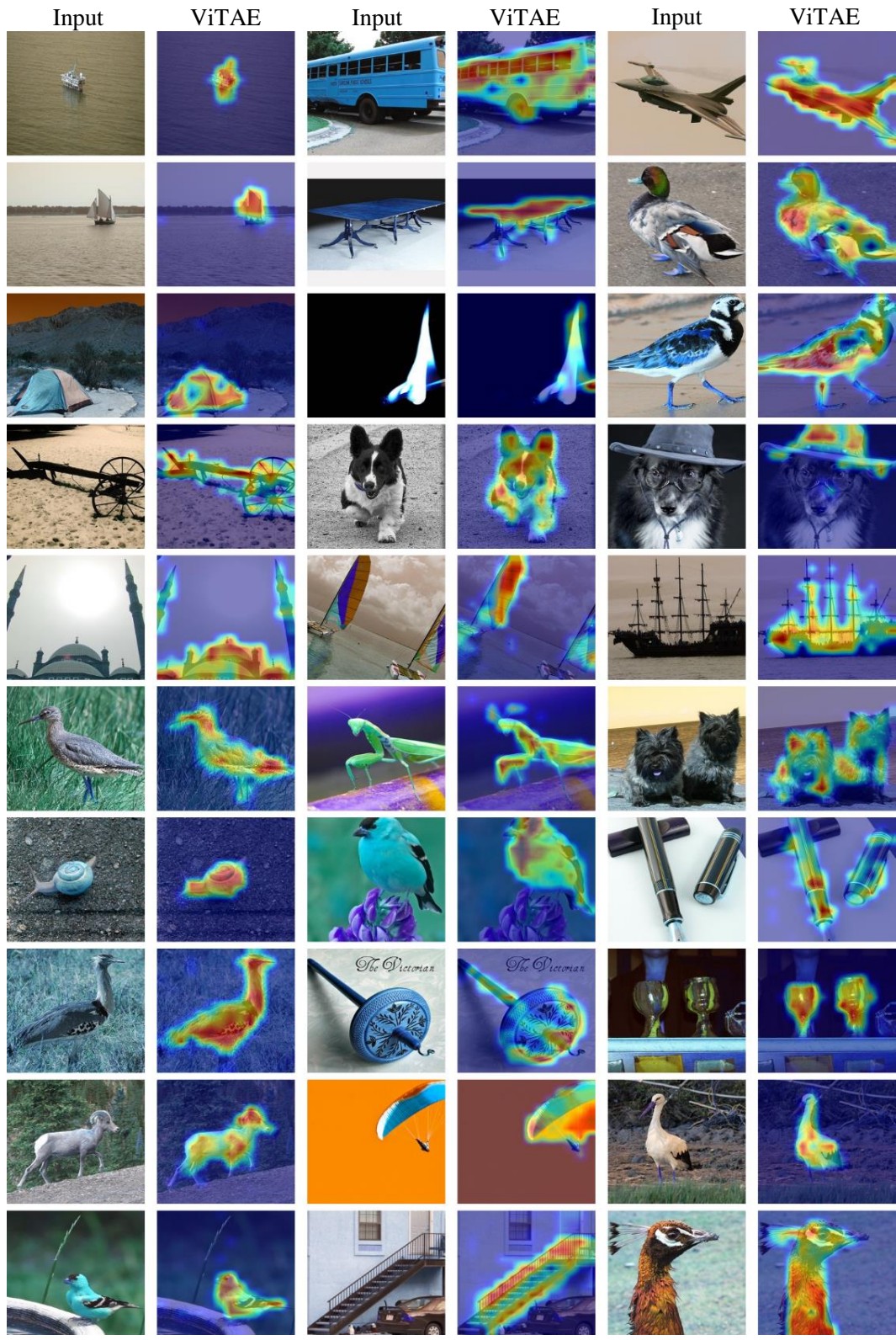

Figure 8: More visual results of ViTAE.

Input ViTAE Input ViTAE Input ViTAE

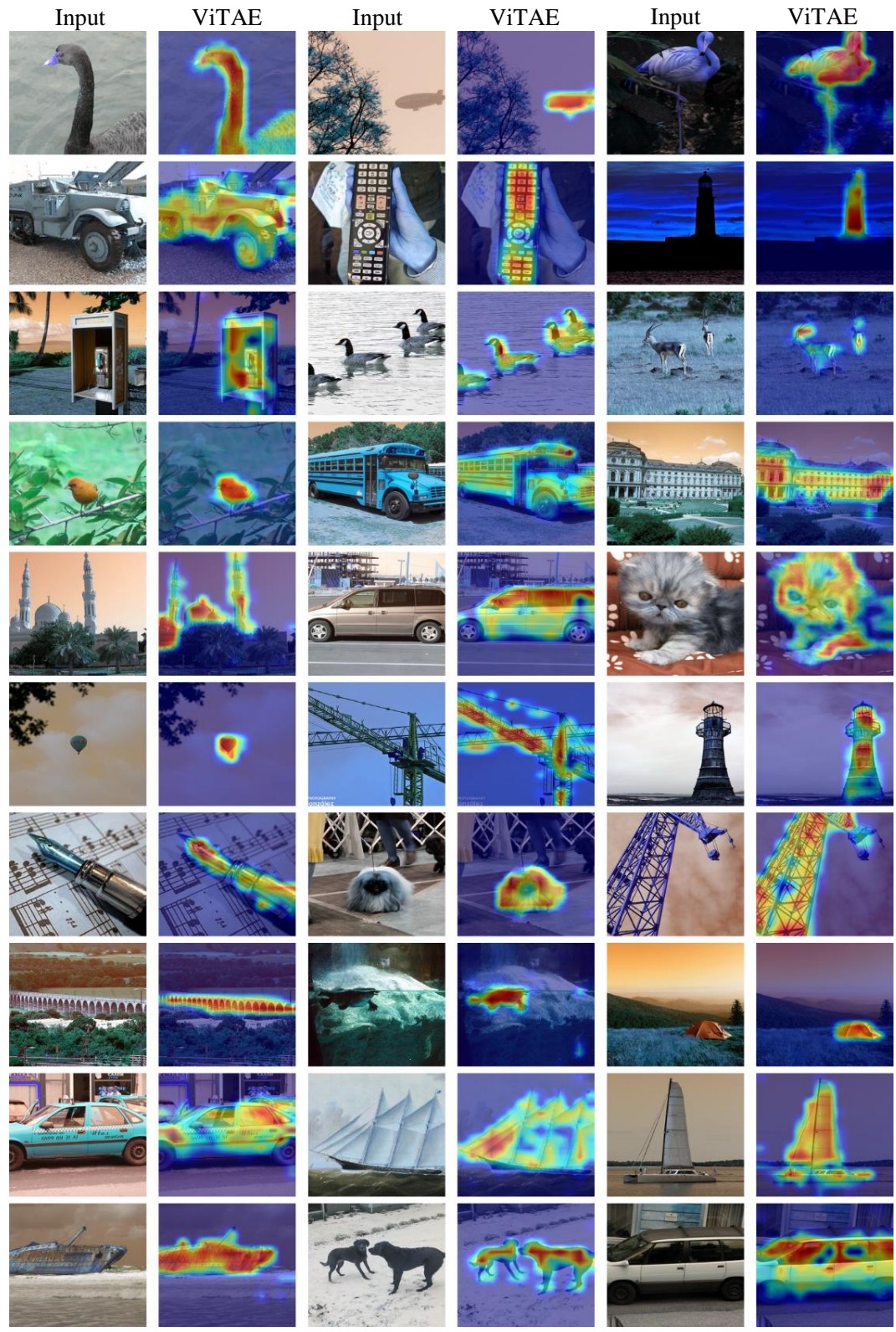

Figure 9: More visual results of ViTAE.

| Input | ViTAE | Input | ViTAE | Input | ViTAE |
|-------|-------|-------|-------|-------|-------|

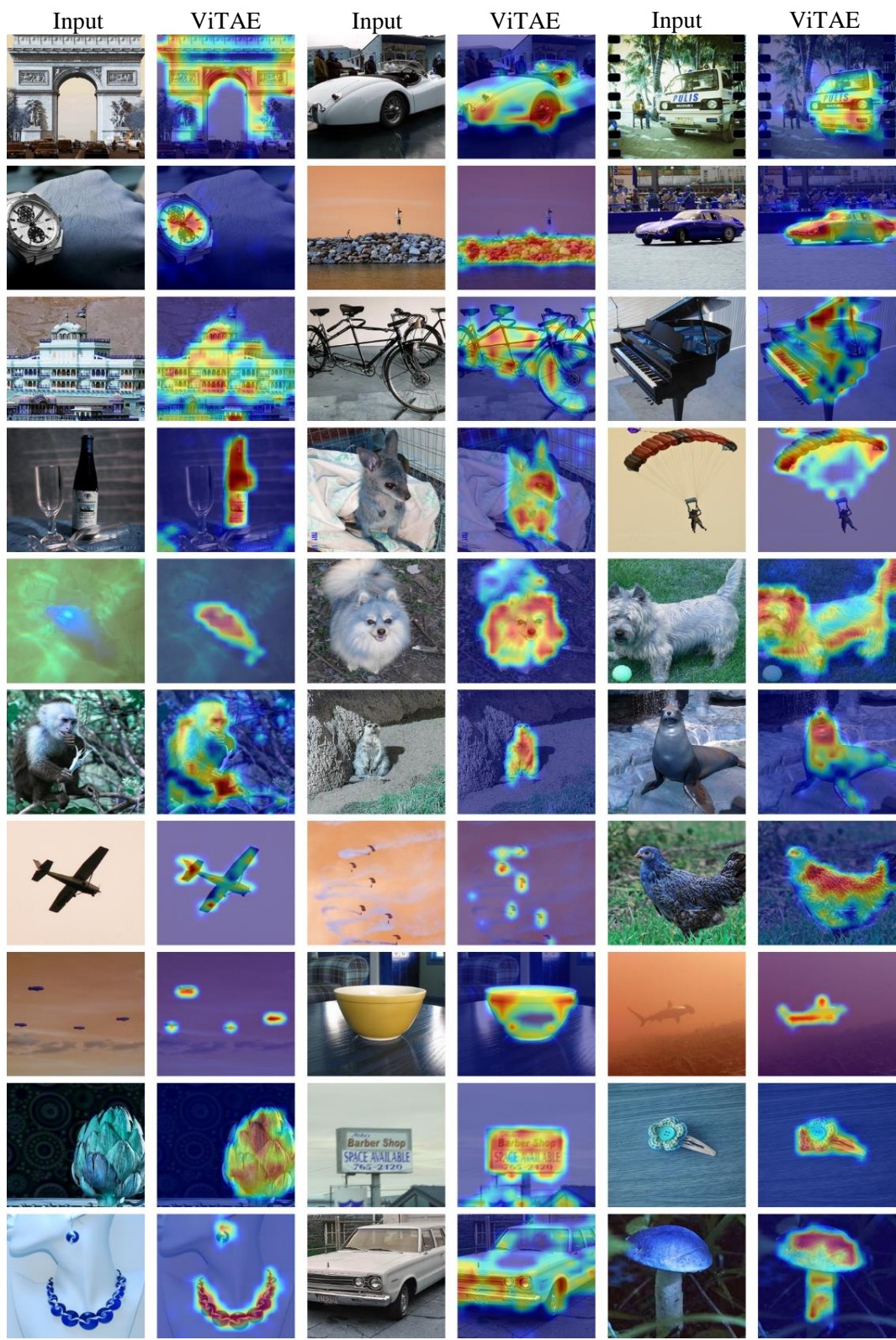

Figure 10: More visual results of ViTAE.

| Input | T2T-ViT | ViTAE | Input | T2T-ViT | ViTAE |
|-------|---------|-------|-------|---------|-------|

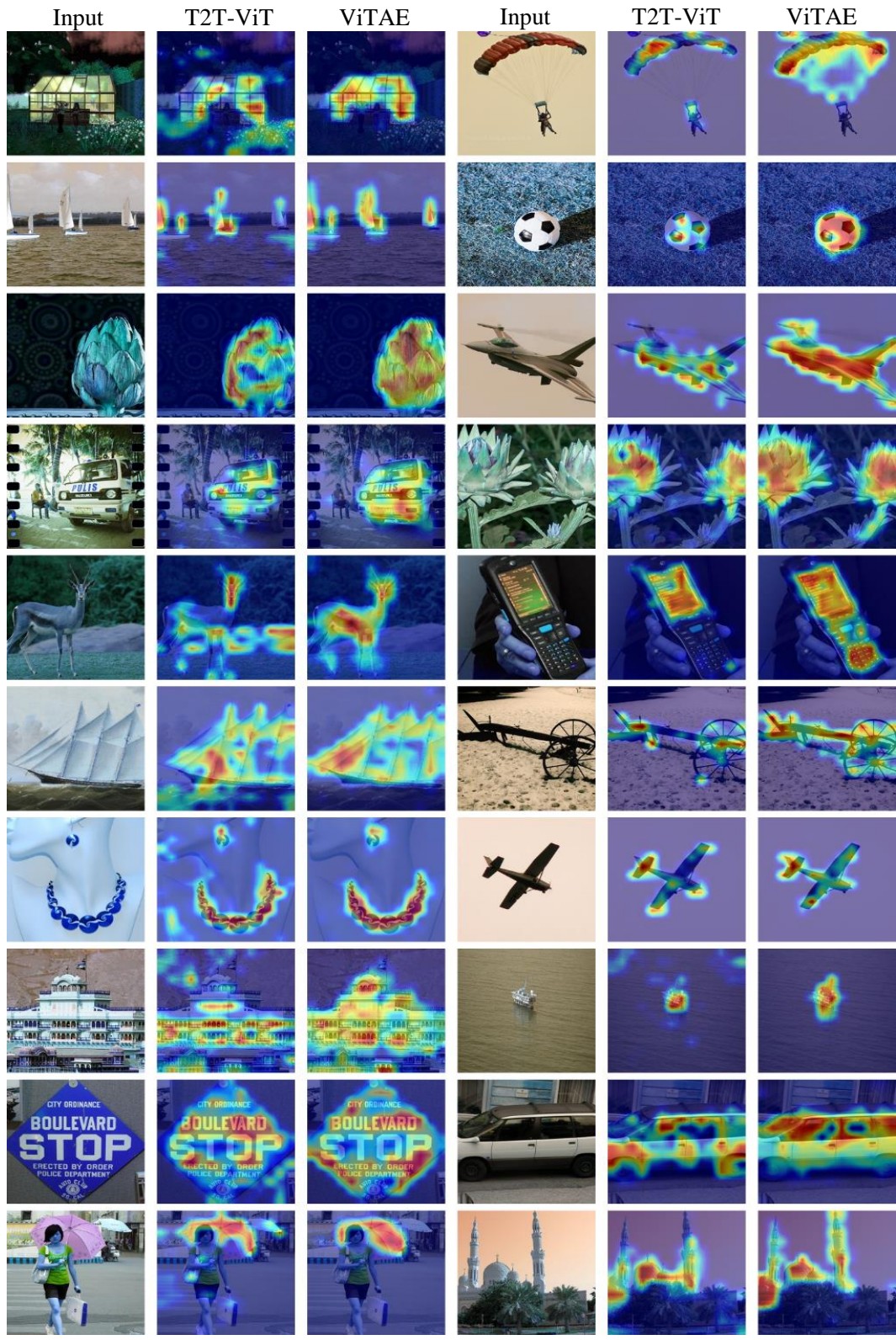

Figure 11: Visual comparison between ViTAE and T2T-ViT.