# OpenReview forum: "ViTAE: Vision Transformer Advanced by Exploring Intrinsic Inductive Bias"
_NeurIPS.cc/2021/Conference — NeurIPS 2021 Poster_

### Official Review · Reviewer_Zi4o · 2021-07-06

**Rating:** 8
**Confidence:** 3

**Summary:**

A new vision transformer architecture VITAE, with a better use of intrinsic inductive bias IB via a new architecture using convolutional dillated layers.
Both scale and locality IB are enabled by that architecture.
Empirical results show that the new method can achieve excellent results with much less parameters than other recent methods.

**Limitations And Societal Impact:**

section 5 is a bit unclear. It is mentioned that the method ViTAE cannot do large size datasets yet. Please discuss better the computational bottlenecks, and best discuss this already in the whole Experiments section 4

**Main Review:**

Overall, it is a good storyline that using the right types of IB can allow more efficient architectures with less parameters.
The paper is very empirical, but seems that the IB does indeed help ViT types of architectures, and this is a good contribution.
However, the paper can benefit from clarity in few points, and also better explanation of the experiments.

- scale invariance and locality IB -- is there an ablation showing how much each contributes to the overall efficiency?

- figure 2 architecture --- reduction cells have self-attention (MHSA) at a relatively large spatial resolution. Does this cost dearly in memory and computational costs -- the ViT only does attention at the short sequence of the final tokens?

- normal cell -- it is not so clear to me how it is different than the usual ViT approach. Are the small changes here really necessary, or the reduction cell + normal ViT would work already very good?

- 3.5 what does "fair comparison" mean? Are the author referring to number of parameters, memory cost, computational runtime? Experiments are done for 300 epochs on V100, but does each epoch cost the same resources for the different methods?

- 4.4. data efficiency -- while the message is clear overall, i am puzzled by the discrepancy of Figure 1 and Table 2. I would expect that at 100% data,  ViT and ViTAE are the same scores as in the table 2, but this is not the case,

**Time Spent Reviewing:**

5

---

> ### Author Response · Authors · 2021-08-10
> **Thanks and response to concerns**
>
> We sincerely thank you for the careful and thoughtful comments. Below we address the key concerns.
>
> *1. Ablation shows how much scale invariance and locality contributes to the overall efficiency.*
>
> **A1:** The ablation study of how much each cell contributes to the overall efficiency is provided in Section 4.3 and Table 3. We are sorry that we do not make it clear in the paper. We train each variant of ViTAE for 100 epochs. The origin model without scale invariance and locality obtains 68.7 Top-1 accuracy. Normal cell only (the locality) obtains 69.9 Top-1 accuracy (the fourth row), and Reduction Cell only (the scale invariance) boosts the accuracy from 68.7 to 71.7 (the second last row). Using both the Reduction Cell and Normal Cell further improves the model's performance to 72.6 accuracy. We will make it clear in the revised version.
>
> *2. Reduction cells have MHSA at a relatively large spatial resolution.*
>
> **A2:** Thanks for your comment. As you mentioned, applying standard multi-head attention to the large spatial resolution may cause unaffordable computation costs. Indeed, we adopt [performer](https://arxiv.org/abs/2009.14794), a popular lightweight attention computation method, to reduce the computation cost in the Reduction cells as [T2T](https://arxiv.org/abs/2101.11986). We are sorry that we did not clarify this implementation detail. We will provide the details in the revised version.
>
> *3. Does Reduction cell + Normal ViT work already good?*
>
> **A3:** Thanks for your comment. The results of using "Reduction cell + Normal ViT" have been provided in the penultimate row in Table 3, where all the options in the normal cell are not selected and only the normal ViT block is used. We are sorry that we do not make it clear. The performance of this model variant achieves 71.7 Top-1 accuracy on ImageNet. Using both the Reduction Cells and Normal Cells results in 72.6 accuracy. These results suggest that convolutions are still required in the normal cells to help transformers focus on what they are good at, e.g., long-range dependencies, and set free transformers from modeling locality. Figure 3 provides empirical evidence where the transformers' attention distances are increased with the help of the introduced convolution layers. We will make it clear in the revised version.
>
> *4. The meaning of fair comparison.*
>
> **A4:** The fair comparison in Section 3.5 means we compare models with comparable parameters using the same training epochs. Hence, the term of fair comparison primarily focuses on the model size and training schedule. We will make it clear in the revised version. Indeed, the computation costs in each epoch of different models are not the same since they have different parameters and Flops. To make it clear, we have tested the training cost of T2T and our ViTAE using NVIDIA Tesla V100. The training of T2T-ViT-7 takes about 12.48 GPU days, while the training of ViTAE-T takes about 13.2 GPU days at the same training settings. We will clarify this term to make it clear in the revised version. Thanks for your comment.
>
> *5. The discrepancy of Figure1 and Table 2.*
>
> **A5:** In the left part of Figure 1, the 20\%, 40\%, and 100\% data means that we train the T2T-ViT and ViTAE using the corresponding portion of ImageNet data, but for equivalent iterations with the same batchsize, i.e., equivalent to the training iterations when training the models using 100\% data for 100 epochs. In this way, the models in all experiments can see the equivalent number of images. In the right part of Figure 1 where the 100\% data are used, we train the model for 100, 200, and 300 epochs, respectively. The results in Table 2 come from the model trained with 100\% data for 300 epochs, which should be the same as the right part in Figure 1. We will make it clear in the revised version.
>
> Besides, we train T2T-ViT and ViTAE using different portions of data for more iterations, i.e., equivalent to the training iterations when training the models using 100\% data for 300 epochs. The results are shown in the following table, where we use 300* to denote the equivalent training schedule when part of data is used.
>
> |  Model | Top-1 Acc | Data Percent | Epochs |
> | :----: | :----: | :----: | :----: |
> | T2T-ViT-7 | 66.7 | 20\% | 300* |
> | ViTAE-T | 69.2 (+2.5) | 20\% | 300* |
> | | | | | |
> | T2T-ViT-7 | 71.1 | 60\% | 300* |
> | ViTAE-T | 74.4 (+3.3) | 60\% | 300* |
> | | | | | |
> | T2T-ViT-7 | 71.7 | 100\% | 300 |
> | ViTAE-T | 75.3 (+3.6) | 100\% | 300 |
> | | | | | |
>
> From the table, we can see that the conclusion of the training efficiency of our ViTAE model still holds. We will revise the description of Figure 1 to make it clear and include the above experimental results in the revised version.
>
> *6. Concerns that ViTAE cannot do large size datasets yet.*
>
> **A6:** Thanks for your comment, and we will add the discussion about large datasets and computation bottleneck in the revised version. ViTAE has not done experiments on large-scale datasets as large-scale datasets, i.e., JFT-300M, are not publicly available now. Besides, we do not have enough computation resources to train the model on large-scale datasets, as training ViT-H using JFT-300M consumes 2.5k TPUv3-core-days. This is still open how inductive bias injected models perform on large-scale datasets. We hope that we can investigate this open problem in the future.
>
> Regards the computation bottleneck, we evaluate the throughout of ViTAE-T and T2T-ViT-7 on a single NVIDIA Tesla V100. The results are shown in the following table.
>
> | Model | InputSize (PX) | Throughout (fps) |
> | :----: | :----: | :----: |
> | T2T-ViT-7 | 224 | 1072 |
> | ViTAE-T | 224 | 1053 |
> | T2T-ViT-7 | 384 | 377 |
> | ViTAE-T | 384 | 360 |
> | T2T-ViT-7 | 768 | 63 |
> | ViTAE-T | 768 | 62 |
> | T2T-ViT-7 | 960 | 32 |
> | ViTAE-T | 960 | 32 |
> | T2T-ViT-7 | 1024| 29 |
> | ViTAE-T | 1024 | 29 |
> | | | |
>
> It can be observed that the speed difference between T2T-ViT-7 and ViTAE-T becomes smaller with the input size increasing. ViTAE adopts parallel structures in each cell, where the transformer branch determines the speed when the input size is large. It is because transformers' computation complexity is $O((HW)^2C)$ while convolutions' computation complexity is $O(K^2HWC^2)$, where $K$ is the kernel size, $C$ is the channel dimension, and $H$, $W$ is the input size. When the input size is large ($HW \gg C$), the computation bottleneck of ViTAE is close to the computation bottleneck of pure transformer models. Thanks for your suggestion again, and we will incorporate the discussion in the revised version.

---

> > ### Comment · Reviewer_Zi4o · 2021-08-11
> > **response to rebuttal**
> >
> > thanks a lot for answering my questions, it addressed well my concerns.

---

### Official Review · Reviewer_eo3B · 2021-07-16

**Rating:** 7
**Confidence:** 5

**Summary:**

The standard vision transformer model lacks the capability to model inductive biases desirable for images e.g., local features encoding, and scale invariance. It, therefore, requires large-scale training data (300 million images), or other variants (e.g., Data Efficient ViT) distilling Knowledge from a pertained CNN. The paper proposes an architecture, which tries to take care of these inductive biases. The architecture comprises of 3 “Reduction cells” at the beginning followed by multiple “Normal Cells”. A reduction cell gets a pyramid effect using dilated convolutions at the beginning which gradually downsamples features. The normal cell doesn’t have any dilated convolutions. Each block has Convolution and attention running in parallel.
The assumption here is that convolution captures local relationships, pyramid structure (via diluted convolutions) captures scale invariance, and attention captures global relationships.


**Limitations And Societal Impact:**

Yes.

**Main Review:**


- A comparison of the proposed model is presented with CNNs and recent ViT variants. The proposed model shows promising results, which are comparable with other competing methods.

- Local features are captured using depth-wise convolutions (e.g., https://arxiv.org/abs/2104.06399), or local attention within a patch (e.g., SwinTransformer). Also, hierarchical feature encoding in a pyramid-like structure is common, and most recent Vision Transformer variants (which I’d categorize as concurrent works) get a spatial feature pyramid by token-merging and projecting the features to a higher dimension, progressively along the hierarchy of the model. These existing works do not nullify the contributions of this paper. The authors just need to clearly state how the existing Vision Transformer variants model scale invariance, and local features; and then justify their proposed approach. In summary, the paper is not the first attempt to introduce inductive biases in Vision Transformers. The paper needs to clearly state that, and change the relevant parts of the paper.

- Capturing the inductive biases (scale invariance+local-features) enables vision transformers to translate well for dense prediction tasks (detection and segmentation). I was expecting empirical evaluations with the proposed architecture used as a backbone for Dense Prediction. For example, these works (which have local-feature-encoding + scale invariance) report results for detection and segmentation.

    - see 4.2 and 4.3 of https://arxiv.org/pdf/2103.14030.pdf
    - see 5.2 and 5.3 of https://arxiv.org/pdf/2102.12122.pdf
    - https://arxiv.org/abs/2106.09681
    - https://arxiv.org/abs/2106.03650
    - https://arxiv.org/abs/2104.06399
    - https://arxiv.org/pdf/2105.03889.pdf

- I am not entirely convinced that convolution is needed in the “normal cell”. Diluted convolutions in the beginning, as well as the normal convolution operation at the initial blocks, should suffice to capture local relationships and scale invariance. The convolutions in the later parts move us away from the original transformer blocks and seem unnecessary to me. What happens if the authors replace the "normal cells" with standard Transformer blocks.


**Time Spent Reviewing:**

4.5

---

> ### Author Response · Authors · 2021-08-10
> **Thanks and response to concerns**
>
> We sincerely thank you for the careful and thoughtful comments. Below we address the key concerns.
>
> *1. Clearly State the relevant work.*
>
> **A1:** We sincerely thank the reviewer for pointing out the concurrent works that introduce inductive bias into transformers. We will revise the relevant part in the revised version and include the discussion about Coat, LocalViT, Swin, PVT, PiT, and the latest works such as XCiT and Shuffle Transformer.
>
> *2. The evaluations on dense prediction tasks.*
>
> **A2:** Thanks for your comment. We will include the evaluation results of our model on dense prediction tasks in the revised version. We stack the reduction cells and normal cells to produce stage-wise features like Swin and evaluate the model on the dense tasks, including ADE-20K for semantic segmentation and Davis for video object segmentation (VOS). Besides, we also provide the results on the other two downstream tasks, i.e., COCO for object detection and human pose estimation. The results are reported with the popular metrics on each task respectively, as shown in the following table (the results of other methods are all from their papers or obtained by following the official codes).
>
> |  Task | Dataset | Backbone   | Method | Params (M) | Metric |
> | :----: | :----: | :----: | :----: | :----: | :----: |
> | Detection | COCO | ResNet | Mask RCNN 1x | 44 | 38.2 (mAP) |
> | Detection | COCO | Swin | Mask RCNN 1x | 48 | 43.7 (mAP) |
> | Detection | COCO | PVT | Mask RCNN 1x | 44 | 40.4 (mAP) |
> | Detection | COCO | Coat | Mask RCNN 1x | 39 | 43.6 (mAP) |
> | Detection | COCO | Conformer | Mask RCNN 1x | 58 | 43.6 (mAP) |
> | **Detection** | **COCO** | **ViTAE**  | **Mask RCNN 1x** | **37** | **44.6 (mAP)** |
> | | | | | | |
> | Segmentation | ADE20K | Swin | UPerNet | 60 | 45.8 (mIoU) |
> | Segmentation | ADE20K | Shuffle | UPerNet | 60 | 47.6  (mIoU) |
> | Segmentation | ADE20K | XCiT | UPerNet | 52 | 46.7 (mIoU) |
> | **Segmentation** | **ADE20K** | **ViTAE** | **UPerNet** | **49** | **47.8 (mIoU)** |
> | | | | | | |
> | Pose | COCO | ResNet | SimpleBaseline | 34 | 71.8 (mAP) |
> | **Pose** | **COCO** | **ViTAE** | **SimpleBaseline** | **27** | **73.7 (mAP)** |
> | | | | | | |
> | VOS | Davis2017 | ResNet | STM | 39 | 81.8 (J&F) |
> | **VOS** | **Davis2017** | **ViTAE** | **STM** | **19** | **82.5 (J&F)** |
> | | | | | | |
>
> It can be concluded that the proposed ViTAE model that introduces inductive bias in reduction cells and normal cells outperforms representative models on downstream dense prediction tasks. We will include these results and more analysis in the revised version.
>
> *3. Replace normal cells with standard transformer blocks.*
>
> **A3:** Thanks for your suggestion. The experiment results of replacing normal cells with standard transformer blocks are available from the 6th row to the 11th row in Table 3 in the paper, where all the options in normal cells are set to false (`x` in the table). We are sorry that we do not make it clear in the paper. As can be seen, the architecture of "reduction cells + standard transformer blocks" achieves 71.1 Top-1 accuracy, while the architecture of "reduction cells + normal cells" achieves 72.6 Top-1 accuracy. These results suggest that convolutions are still required in the normal cells to help transformers focus on what they are good at, e.g., long-range dependencies, and set free transformers from modeling locality. Figure 3 provides empirical evidence where the transformers' attention distances are increased with the help of the introduced convolution layers. We will make it clear in the revised version.

---

> > ### Comment · Reviewer_eo3B · 2021-08-31
> > **Thanks for the response**
> >
> > I thank the authors for the response and additional evaluations on the dense prediction. I will upgrade my rating based upon this.

---

### Official Review · Reviewer_dFPw · 2021-07-20

**Rating:** 8
**Confidence:** 4

**Summary:**

The paper propose combining convolution layers, which capture relevant intrinsic biases of images, with attention layers, which can can capture long-distance relations, to learn better models for image classification.

Additionally, the paper analyze different ways to combine lessons from CNNs (residual layers, BN, multi-scale, spatial reduction) with the latest advancements in visual transformers to obtain more powerful models, that have lower number of parameters but can compete with state-of-the-art models.

**Ethical Concerns:**

As visual models get more and more accurate the possibilities of misuse grows, so mitigation strategies and careful design of experiments and applications are needed.

**Limitations And Societal Impact:**

The paragraph about Broad impacts in section 6 is a just a generic statement, and it doesn't reflect on the possible negative consequences of having more precise visual models.

**Main Review:**

# Originality:
The paper proposes a novel combination of convolution networks with transformer networks, that seems to get the best of both worlds. Implicit locality, multi-scale, and parameter sharing of CNNs with the long-range learnable relations of T2Ts.

This work builds of previous works, but finds a novel architecture that performs better and requires much less parameters than alternative methods.

I would recommend taking a look at "Early Convolutions Help Transformers See Better" concurrent work which also combine convolution with transformers and obtain very good results with stable training.

# Quality:
Overall, the paper is technically sound, the experiments, comparisons and ablation studies increase the confidence that the proposed architecture provides significant improvements, and disentangle the different contributions of the design choices.

A comparison with a network that replaces the Reduction part of the network with just multiple convolutions that reduce the input size in a similar fashion would helpful to understand the need of attention layers in the reduction cells.


# Clarity:
The paper is clear and well organized, there are some small typos (see minor comments below), which allow the reader to get good understanding of the proposed architecture and enough information to reproduce the results.

It's unclear what real Top-1 accuracy means.

# Significance:
The specific way of mixing CNNs and transformers seems to provide substantial value while at the same time reducing the number of parameters and maintaining the computational requirements.

The papers provided improvements on ImageNet results compared with a broad set of state-of-art methods.

If the improvements in ImageNet classification task could be translated to COCO detection task, where the multi-scale and locality properties are critical, it would significantly increase the significance of the paper.


# Minor:
Line 288 "RCs and RCs" -> "RCs and NCs"

In appendix Figure S2 caption there are two (c).

# After further analysis and extra experiments provided by the authors updated score to 8.

**Time Spent Reviewing:**

4

---

> ### Author Response · Authors · 2021-08-10
> **Thanks and response to concerns**
>
> We sincerely thank you for the careful and thoughtful comments. Below we address the key concerns.
>
> *1. The need for attention layers in the reduction cells.*
>
> **A1:** Thanks for your comment. We implement the reduction cells by removing the transformer layers and keeping other parts the same as the original ViTAE model. The Top-1 accuracy of this variant is 71.3 accuracy (compared with 72.6 accuracy with the original design). It shows that modeling long-range dependencies is also helpful during the downsampling stage, since the model can learn more representative features by comparing and collecting similar patterns from a nonlocal neighborhood. A hybrid block that can simultaneously model both short- and long-range dependencies are beneficial for vision tasks, as claimed in the paper.
>
> *2. The real Top-1 accuracy's meaning.*
>
> **A2:** Sorry for the missing explanation of this metric. The real Top-1 accuracy means we evaluate the accuracy on the ImageNet with labels provided by the paper ["Are we done with ImageNet?"](https://arxiv.org/pdf/2006.07159.pdf) as done in previous works ([ViT](https://arxiv.org/pdf/2010.11929.pdf), [DeiT](https://arxiv.org/pdf/2012.12877.pdf), etc.). The refined labels remedy the errors in origin labels. We will clarify this metric to make the paper more readable.
>
> *3. Performance of ViTAE model on downstream tasks.*
>
> **A3:** Thanks for your suggestion. We will include the results of ViTAE on downstream tasks in the revised version. We stack the reduction cells and normal cells to produce stage-wise features like Swin and evaluate their efficiency on the downstream tasks, including COCO for object detection and human pose estimation, ADE-20K for semantic segmentation, and Davis for video object segmentation (VOS). The results are reported with the popular metrics on each task respectively, as shown in the following table (the results of other methods are all from their papers or obtained by following the official codes).
>
> |  Task | Dataset | Backbone   | Method | Params (M) | Metric |
> | :----: | :----: | :----: | :----: | :----: | :----: |
> | Detection | COCO | ResNet | Mask RCNN 1x | 44 | 38.2 (mAP) |
> | Detection | COCO | Swin | Mask RCNN 1x | 48 | 43.7 (mAP) |
> | Detection | COCO | PVT | Mask RCNN 1x | 44 | 40.4 (mAP) |
> | Detection | COCO | Coat | Mask RCNN 1x | 39 | 43.6 (mAP) |
> | Detection | COCO | Conformer | Mask RCNN 1x | 58 | 43.6 (mAP) |
> | **Detection** | **COCO** | **ViTAE**  | **Mask RCNN 1x** | **37** | **44.6 (mAP)** |
> | | | | | | |
> | Segmentation | ADE20K | Swin | UPerNet | 60 | 45.8 (mIoU) |
> | Segmentation | ADE20K | Shuffle | UPerNet | 60 | 47.6 ( mIoU) |
> | Segmentation | ADE20K | XCiT | UPerNet | 52 | 46.7 (mIoU) |
> | **Segmentation** | **ADE20K** | **ViTAE** | **UPerNet** | **49** | **47.8 (mIoU)** |
> | | | | | | |
> | Pose | COCO | ResNet | SimpleBaseline | 34 | 71.8 (mAP) |
> | **Pose** | **COCO** | **ViTAE** | **SimpleBaseline** | **27** | **73.7 (mAP)** |
> | | | | | | |
> | VOS | Davis2017 | ResNet | STM | 39 | 81.8 (J&F) |
> | **VOS** | **Davis2017** | **ViTAE** | **STM** | **19** | **82.5 (J&F)** |
> | | | | | | |
>
> Swin, PVT, Coat, Conformer, XCiT, and Shuffle are different transformer-based methods. It can be observed that our ViTAE model outperforms SOTA models on various downstream tasks even if it is smaller in model size, confirming the value of the proposed ViTAE architecture.
>
> *4. Minor typos.*
>
> **A4:** Thanks for your comment. We will carefully proofread the whole paper. We will also review the concurrent work 'Early Convolutions Help Transformers See Better' in the revised version as well as discuss more detailed broad impacts.

---

### Official Review · Reviewer_D6zB · 2021-07-26

**Rating:** 6
**Confidence:** 3

**Summary:**

This work incorporates two architecture changes which induce the inductive biases of 1) locality and 2) scale-invariance. It does so though a sequence of Reduction Cells followed by Normal Cells. In doing so, the authors demonstrate comparable performance to strong CNN baselines and improvements over Transformer-based ImageNet models -- the top performing ViTAE model achieves a 83.0% Top-1 accuracy.

**Limitations And Societal Impact:**

Yes.

**Main Review:**

Notes:
* Writing nit: if you’re proposing a method, you don’t need to also say it’s novel.
* Writing nit: odd opening sentence (“domination trend” and an emphasis that Transformers perform well in NLP due to an ability to model long-range dependencies, not really a focus in this work).
* Thanks for presenting the Ablation Study in Table 3. The architecture details were complicated so this helps the reader attribute significance to different pieces.


Questions:
* ResNet-RS (Bello et al., 2021) notes that FLOPs can often be a poor predictor of step-time on modern accelerators due to how well certain operations are supported. Further, parameters can be a bad proxy for memory usage. Can the authors comment on the wall-clock speed of ViTAE versus other baselines? Also, Table 2 might include the ResNet-RS variants which are Pareto optimal and perform better than EfficientNet.
* How do the authors envision the future of vision architecture design and how would they motivate this to a vision researcher? If larger-scale data is available, low-inductive bias models like ViT have shown great promise, but if the data is more limited, high-inductive bias models like ResNets/EfficientNets continue to produce Pareto-optimal models. The context of how this contribution fits in at a high level is what prevented a higher initial score. Could the authors comment?


**Time Spent Reviewing:**

4

---

> ### Author Response · Authors · 2021-08-10
> **Thanks and response to concerns**
>
> We sincerely thank you for the careful and thoughtful comments. Below we address the key concerns.
>
> *1. Writing nit.*
>
> **A1:** Thanks for your suggestion, and we will improve the writings accordingly in the revised version.
>
> *2. The wall-clock speed of ViTAE versus other baselines.*
>
> **A2:** Thanks for your comment. We will include ResNet-RS variants in Table 2. Regarding the wall-clock speed, we have tested the variants of the ViTAE model and the classic transformer model (T2T) on a single NVIDIA Tesla V100. The results are summarized in the following table. Thanks to the parallel branches adopted in both RCs and NCs, the ViTAE variants only suffer a negligible loss in speed. Specifically, our ViTAE-T model outperforms T2T-ViT-7 by 3.6 Top-1 accuracy on ImageNet but matches the throughput speed of T2T-ViT-7, i.e., 1053 vs. 1072 images per second. Similarly, the throughput speed of our ViTAE-S is slightly slower than T2T-ViT-14, i.e., 535 vs. 573, while it achieves a gain of 0.5 accuracy. Our model has a faster throughput speed when comparing different models under the same Top-1 accuracy, i.e., 82.0 of ViTAE-S and 81.9 of T2T-ViT-19. We will report these results in Table 2 in the revised version of the paper.
>
> |  Model | Params | Top-1 Acc | Throughput (images/per second) |
> | :----: | :----: | :----: | :----: |
> | T2T-ViT-7 | 4.3 | 71.7 | 1072 |
> | ViTAE-T | 4.8 | 75.3 | 1053 |
> | | | | |
> | T2T-ViT-14 | 21.5 | 81.5 | 573 |
> | T2T-ViT-19 | 39.2 | 81.9 | 382 |
> | ViTAE-S | 23.6 | 82.0 | 535 |
> | | | |
>
> *3. How do the authors envision the future of vision architecture design and how would they motivate this to a vision researcher? What is the real contribution of the paper at a high level for the future vision architectures?*
>
> **A3:** Thanks for your comments. We agree with you that if larger-scale data is available, low-inductive bias models like ViT have shown great promise. Still, if the data is more limited, high-inductive bias models like ResNets/EfficientNets continue to produce Pareto-optimal models. Nevertheless, insufficient computation resources and the limited scale of data slows the adoption of low-inductive bias models. Firstly, it is very difficult to collect and annotate large-scale data regarding privacy, expenses, etc., e.g., hundreds of millions of images in JFT-300M, especially for specific areas like medical image analysis. Secondly, training a large model using hundreds of millions of images requires a large computation infrastructure. For example, training ViT-H on JFT requires 2.5k TPUv3-core-days, limiting the practical application potentials of transformers.
>
> Therefore, we believe that improving the training efficiency and getting better performance by training from scratch on middle-size datasets also matters for the computer vision community and is of practical significance. In this paper, we make an attempt towards this direction by proposing the ViTAE model, which is devised from the perspective of incorporating intrinsic inductive bias from CNNs into vision transformers with low-inductive bias. Such a design (with other concurrent works) and its promising results show the importance of inductive bias in improving the training efficiency on middle-size datasets, e.g., ImageNet. These works provide alternatives to the low-inductive bias transformers when there are no large computation infrastructure and sufficient training data. Also, they provide a new perspective to the community to explore the potential of transformers.
>
> Besides, it remains open about the behavior of such models with injected inductive bias on large-scale datasets such as JFT-300M. Nevertheless, we observe that introducing the convolutions helps the transformer layer focus on long-range dependencies modeling, as shown in Figure 3 in the paper. We hope that we can investigate the role of inductive bias, e.g., the one from convolutions, in the future and make an attempt to address this challenge. Thank you again for your inspiring comments.

---

### Author Response · Authors · 2021-08-10
**Thanks for the reviews**

We sincerely thank the reviewers for their thoughtful reviews. We are encouraged that the reviewers appreciate the SOTA performance of ViTAE on classification (Reviewer D6zB, dFPw, eo3B, Zi4o), a novel architecture (Reviewer dFPw, Zi4o), and the efficiency of the two introduced inductive biases (Reviewer D6zB, dFPw, eo3B, Zi4o).

We provide detailed responses to each reviewer respectively and promise will incorporate all feedbacks in the revised version.

---

### Decision · Program_Chairs · 2021-09-28

**Decision:**

Accept (Poster)

**Comment:**

This work proposes a new method for combining convolution layers with attention in computer vision problems by allowing the attention layers to focus on capturing long range correlation structures. In addition, this work analyzes multiple dimensions for how to combine convolutional networks with visual transformers in order to achieve favorable performance in terms of computer vs performance. All of the reviewers were impressed by the soundness of the experiments, the clarity of the exposition and the significance of the results. For all of these reasons, this paper will be accepted.


**Consistency Experiment:**

NeurIPS has a long history of experimentation. In 2014, NeurIPS ran an experiment in which 10% of submissions were reviewed by two independent committees to quantify the randomness in the review process. This year, we repeated a variant of this experiment to see how the quality of the review process has changed over time.  This paper was part of the experiment and was therefore assigned to two committees (consisting of reviewers, an Area Chair, and a Senior Area Chair) that reached independent decisions.  If both committees made the same recommendation, this recommendation was followed. If a single committee recommended acceptance, the paper was accepted (with the exception of a few cases in which the other committee identified what we considered a fatal flaw, e.g., an error in a key result).

This copy’s committee reached the following decision: **Accept (Poster)**

The other committee assigned to the paper recommended **Reject**.  You can find the other set of reviews, along with any follow up discussion with the authors here:
https://openreview.net/forum?id=_WnAQKse_uK